# How Does Reasoning Flow? Tracing Attention-Induced Information Flow for Targeted RL in LLMs

Zhichen Dong [* 1 2]  Yang Li [* 1 2 3]  Yuhan Sun [1 3]  Weixun Wang [2]  Yijia Luo [2]  Zinian Peng [1]
Wenbo Su [2]  Cheng Yu [2]  Bo Zheng [2]  Junchi Yan [1 3]

## Abstract

Token-level credit assignment remains a key obstacle for reinforcement learning (RL) in large language models (LLMs), where RL recipes typically treat all tokens equally, failing to distinguish decisive reasoning steps from routine formatting or fluent filler. Recent attempts leverage model-internal signals to assign finer-grained credit, but these are often point-wise heuristics that ignore the global structure of information propagation. We propose FlowTracer, an RL framework that traces *answer-targeted reasoning flow* on an attention-induced directed acyclic graph in which nodes correspond to tokens and edge capacities come from aggregated attention weights and derives token credit from this global structure. The edge capacities are reweighted to retain only the influence that can reach the answer region, while enforcing local flow conservation so intermediate tokens neither lose nor gain effective mass due to path length or irrelevant branches. On this graph, FlowTracer extracts an information-flow backbone connecting the question to the answer and scores tokens by flow throughput, revealing high-impact hubs and aggregation checkpoints that mediate long-range dependencies. These derived importances are used to shape token-level rewards, enabling learning signals to focus precisely on the tokens that route information toward (or away from) correct answers and delivering consistent performance gains across a range of reasoning tasks.

---

[*]Equal contribution  [1]Shanghai Jiao Tong University  [2]Alibaba Group  [3]Shanghai Innovation Institute. Correspondence to: Junchi Yan <yanjunchi@sjtu.edu.cn>, Wenbo Su <vincent.swb@alibaba-inc.com>.

*Proceedings of the 43rd International Conference on Machine Learning*, Seoul, South Korea. PMLR 306, 2026. Copyright 2026 by the author(s).

## 1 Introduction

Reinforcement learning (RL) has become an important tool for training and aligning large language models (LLMs), and it has shown particular promise in eliciting and strengthening step-by-step reasoning for complex tasks (Christiano et al., 2017; Jaech et al., 2024; Guo et al., 2025; Team et al., 2025b). Among RL approaches, reinforcement learning with verifiable rewards (RLVR) offers a practical and scalable recipe when automatic checkers are available, providing reliable training signals from deterministic evaluators (Shao et al., 2024; Lambert et al., 2024). This paradigm has driven substantial progress in mathematical problem solving (Xin et al., 2025; Wang et al., 2024; Long et al., 2025), code generation (Hui et al., 2024; Yang et al., 2025a), and more recently agentic settings (Yang et al., 2024; Team et al., 2025a; Wang et al., 2025d), where sequences of intermediate decisions unfold over long horizons and task success is precisely defined and automatically verifiable.

Despite these gains, a persistent limitation of RL for LLMs is token-level credit assignment. Autoregressive generation produces long trajectories with sparse, delayed supervision (e.g., correctness judged only at the end), so attributing success or failure to individual tokens is inherently challenging. While classical RL tools such as generalized advantage estimation (GAE) (Schulman et al., 2015) can in principle yield token-wise learning signals, they rely on accurate state-value estimates. In practice, estimating token-level value from the outside, i.e., based only on observed text states, is difficult in rich linguistic contexts, which makes attribution noisy and unstable. This often pushes methods toward coarse approximation designs that effectively weight tokens uniformly (Shao et al., 2024; Zheng et al., 2025), obscuring the difference between pivotal reasoning steps and incidental wording. On the other hand, recent works show that the model's own internal dynamics could provide additional reference signals for credit assignment (Wang et al., 2025b; Cui et al., 2025; Li et al., 2025d), e.g., indicators of when information is being accumulated versus when the model is uncertain or confused. However, existing approaches typically operationalize such cues via point-wise heuristics (e.g., entropy (Wang et al., 2025b;a), attention

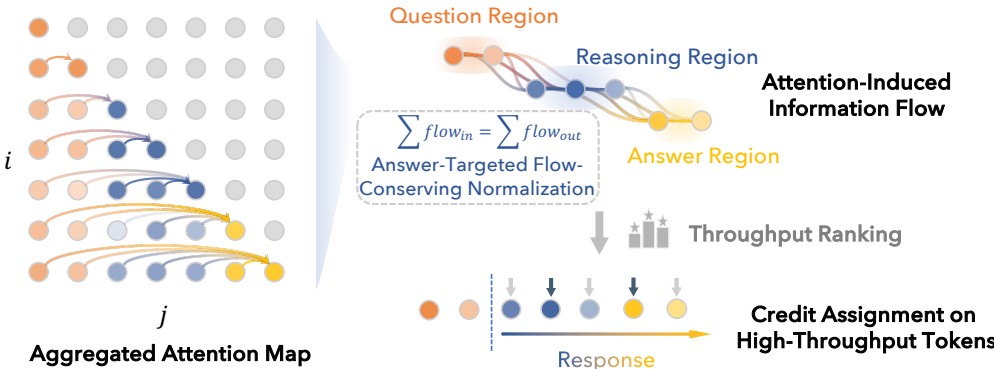

*Figure 1.* **Overview of FlowTracer.** We build an attention-induced token graph, condition it on the answer to retain only routes that support the prediction, and normalize it to ensure locally consistent flow. Injecting flow from a super-source over the prompt to a super-sink over the answer yields a backbone of dominant multi-hop information paths. Token throughput on this backbone identifies key routing hubs for token-level credit assignment and reward shaping.

statistics (Bogdan et al., 2025; Li et al., 2025d)) and largely overlook the global structure of how information propagates and is transformed across the full sequence.

We argue that resolving token-level credit assignment requires answering a more fundamental question: *How does reasoning flow from the prompt to the final answer inside an LLM?* Rather than scoring tokens in isolation, we seek a structured, system-level characterization of how information is routed through intermediate tokens and long-range dependencies. Attention provides an explicit interaction graph among tokens, but raw attention weights alone do not directly reveal which multi-hop routes constitute the dominant backbone that carries task-relevant information. This motivates a graph-theoretic approach that can extract globally consistent paths and identify the true transit hubs that mediate prompt-to-answer transmission.

To this end, we propose FlowTracer, a graph-based credit assignment method that extracts an *answer-targeted reasoning backbone* from the model's attention structure and uses it to guide RL updates. We view the token sequence as an attention-induced directed acyclic graph (DAG), where each node is a token and each directed edge carries a nonnegative capacity derived from aggregated attention weights, interpreted as the strength of potential information transfer. Our goal is to quantify, for each token, how much of its influence is actually routed toward the answer region, rather than merely being locally salient.

A direct use of raw attention is insufficient for this purpose. Attention graphs contain numerous branches that never contribute to the final answer, and naive propagation on such graphs causes influence to be diluted along long paths or absorbed by irrelevant subtrees. As a result, early but decisive premises can be systematically under-credited, while late-stage tokens near the answer may receive disproportionate weight simply due to proximity. To avoid these

structural biases, FlowTracer introduces an explicit *answer-conditioning* step on the attention graph: we first define an answer region (e.g., the final answer span) and compute a global reachability potential that measures how much downstream influence from each token can ultimately reach the answer. We then *reweight* outgoing edge capacities to keep only the answer-relevant portion of influence and to satisfy a local flow-conservation property, ensuring that intermediate tokens neither lose nor gain effective mass due to path length, fan-out, or irrelevant branches. Tokens with zero answer-reachability are naturally filtered out, yielding a cleaned, target-conditioned graph that focuses analysis on the effective reasoning substructure.

On this target-conditioned graph, FlowTracer performs a flow analysis between the prompt and the answer. By injecting unit flow from a super-source connected to prompt tokens and collecting flow at a super-sink connected to answer tokens, we recover an information-flow backbone that highlights the dominant multi-hop routes supporting the final prediction. The resulting token throughputs discover the reasoning pattern where high-throughput tokens act as transit hubs or aggregation checkpoints that mediate long-range dependencies (e.g., periodic consolidation near clause boundaries, repeated symbols, or variables that serve as cross-step anchors). By applying the resulting token throughputs, we plug the globally consistent notion of credits into RLVR via token-level reward shaping and loss reweighting, amplifying learning signals on high-impact routing tokens and suppressing updates on low-impact filler, thereby improving both learning efficiency and reasoning performance.

## 2 Related Work

**Deriving Optimization Signals From LLM Internal Dynamics.** Generative modeling has shown promise in broad scenarios (Li et al., 2026; 2025e; Chen et al., 2026; Wang

et al., 2026). While traditional optimization treats LLMs as black-box generators for end-to-end optimization, recent studies exploit internal computational processes to extract fine-grained signals (Zou et al., 2023; Chen et al., 2024; Dong et al., 2025). Research identifies specific components crucial for information handling, including task-specific attention heads (Cabannes et al., 2024; Bertolazzi et al., 2025), context-aggregating receiver heads (Ren et al., 2024; Zheng et al., 2024), specialized functional layers (Dumitru et al., 2024), and steering directions in representation space (Burns et al., 2022; Venhoff et al., 2025). These insights enable LLM augmentation via representation editing (Hernandez et al., 2023; Turner et al., 2023), side-route classifiers (Li et al., 2022; Belrose et al., 2023; Ji et al., 2024), and component-focused training (Zhao et al., 2025). Beyond static analysis, dynamic methods examine information propagation via attention (Geva et al., 2023; Bogdan et al., 2025). This reveals internal traits such as factual association (Geva et al., 2023; Mohebbi et al., 2023), multi-path calculation (Dutta et al., 2024; Ameisen et al., 2025) and critical reasoning steps (Bogdan et al., 2025; Minegishi et al., 2025; Qian et al., 2025). However, raw attention is noisy and limited to single-step, point-wise influence. To this end, we apply a Doob-h-like transform to isolate the answer-relevant multi-hop reasoning backbone, yielding a robust signal to guide optimization. This view is also related to recent label-repurposing ideas, in which ground-truth labels are treated not merely as loss targets but as informative references that guide prediction learning (Li et al., 2025f).

**Credit Assignment for RL in LLMs.** RL is standard for LLM post-training (Ziegler et al., 2019; Ouyang et al., 2022; Achiam et al., 2023; Lu et al., 2025), yet effective credit assignment remains an evolving challenge (Lin et al., 2024; Vassoyan et al., 2025; Liu et al., 2026; Dong et al., 2024). Off-policy RL (Rafailov et al., 2023; Meng et al., 2024) aligns probabilities, while more prevailing on-policy RL relies on sparse outcome rewards (Bai et al., 2022). Explicit step-wise supervision (e.g., PRMs (Lightman et al., 2023) or MCTS (Guan et al., 2025)) mitigates sparsity but faces reward hacking and efficiency bottlenecks (Cheng et al., 2025b). With the emergence of RLVR (Shao et al., 2024; Lambert et al., 2024), Group Relative Policy Optimization (GRPO) (Xin et al., 2025) bypasses these limitations by using group advantage as an implicit signal to induce reasoning behaviors (Yu et al., 2025; Yang et al., 2025a). Yet, this approach distributes credit uniformly, failing to distinguish critical steps from fillers (Gandhi et al., 2025; Li et al., 2025a). Recent works improve this with signals like entropy (Wang et al., 2025b;a; Cheng et al., 2025a; Cui et al., 2025), confidence or correlation (Li et al., 2025c; Zhou et al., 2024b; Nie et al., 2025; Zhou et al., 2024a), gradients (Green et al., 2025; Li et al., 2025b) and attention (Li et al., 2025d), but these remain point-wise and

ignore inter-token relationships. Our method models multi-hop influence within the reasoning flow to identify a token's true contribution, yielding a more precise credit assignment.

# 3 Methodology

We present FlowTracer, a principled framework for token-level credit assignment in reinforcement learning (RL) for large language models (LLMs). Our approach leverages the global structure of attention-induced information propagation to identify which tokens genuinely contribute to the final answer. The method rests on three core ideas: (1) modeling reasoning as influence flow over a directed acyclic graph (DAG) induced by attention; (2) enforcing answer-targeted flow conservation through a Doob-h-like reweighting, followed by forward propagation to compute token-level throughput; and (3) using the resulting flow throughput to drive fine-grained, non-uniform policy updates. Crucially, we interpret attention weights as non-negative *influence couplings* and exploit their algebraic structure to isolate only those paths that meaningfully contribute to the final output.

## 3.1 Answer-Targeted Influence Flow on Attention-Induced DAGs

We formalize reasoning as a linear influence propagation process over an attention-induced directed acyclic graph (DAG), and compute a conserved flow that isolates influence paths that ultimately reach the answer. This consists of three stages: (1) constructing a raw influence graph from attention, (2) reweighting edges via a Doob-h-like transform to enforce answer-targeted conservation, and (3) propagating unit source flow forward to obtain token-level credit.

**Raw Influence Graph from Attention.** Given an input-output sequence $(x_1, \ldots, x_T)$ generated by an LLM, we construct a time-ordered DAG $\mathcal{G} = (V, E)$ where each token $x_i$ corresponds to a node $v_i \in V$, and a directed edge $E_{i \to k}$ exists for all $i < k$. The edge weight $W_{ik} \geq 0$ is derived from aggregated attention scores (e.g., mean over attention layers and heads) $a(x_k, x_i)$:

$$W_{ik} \coloneqq a(x_k, x_i) \geq 0, \tag{1}$$

We interpret $W_{ik}$ as the *local influence coupling strength*: when one unit of influence departs from $x_i$, a fraction $W_{ik}$ is absorbed by $x_k$ and may be further propagated. Critically, we do not require $\sum_{k>i} W_{ik} = 1$; thus, the out-degree sum may exceed 1 (parallel broadcasting) or fall below 1 (signal attenuation), and $W$ defines a linear operator instead of a stochastic kernel.

While attention weights provide a natural signal for inter-token dependencies, using the raw attention graph $W$ directly for credit propagation suffers from two critical issues

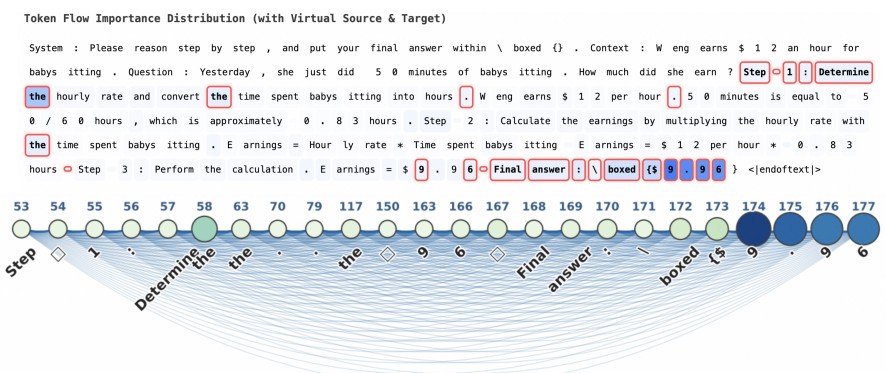

*Figure 2.* Answer-targeted token flow importance in Qwen3-4B. By modeling the generation as a DAG, we trace the influence flow from the prompt to the final answer. Darker nodes indicate higher flow throughput, identifying a reasoning backbone of decisive tokens (e.g., "9.96") over routine filler. This enables finer-grained token-level credit assignment for targeted RL.

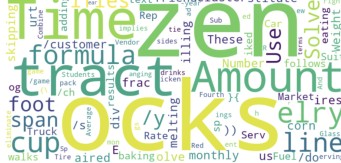

*(a)* High-flow tokens.

*(b)* Low-flow tokens.

*Figure 3.* Word cloud of high-flow and low-flow tokens.

that undermine structured reasoning analysis: First, **attention violates local flow conservation**. Standard attention weights are normalized over *source tokens* for each target, i.e., $\sum_{i<k} W_{ik} = 1$ (in-degree normalization). However, the out-degree sum $\sum_{k>i} W_{ik}$ is generally *not equal to 1*; it can be arbitrarily large (if $x_i$ attends broadly) or small (if $x_i$ is ignored). Consequently, when influence flows forward from a node, it may be amplified or attenuated purely due to graph topology, not semantic importance. Second, and more fundamentally, **the raw graph contains extensive answer-irrelevant substructures**. A large fraction of attention flows into filler tokens, formatting markers, or intermediate hypotheses that are later discarded, i.e., paths that terminate before contributing to the final answer. Propagating credit through such dead-end branches leads to systematic underestimation of early critical premises (due to exponential decay over spurious paths) and overemphasis on tokens near the answer that merely restate conclusions.

**Doob-h-Like Reweighting for Effective Influence.** To resolve both issues, we seek a reweighted graph $W'$ that (1) enforces *local flow conservation* ($\sum_{k>i} W'_{ik} = 1$) and (2) restricts propagation to only those paths that ultimately reach the answer. We achieve this via a *Doob-h-like reweighting*, where $h(i)$ denotes the total influence from node $i$ that successfully reaches the answer. We introduce a virtual sink node $s$ connected from all answer tokens $\mathcal{A}$ and then define a potential function $h(i)$ indicating the *effective reachability* to the answer as the total influence-weighted path sum from node $i$ to $s$:

$$h(s) = 1, \quad h(i) = \sum_{k>i} W_{ik} h(k), \quad \forall i \notin \mathcal{A}. \quad (2)$$

Then we set

$$W'_{ik} := \frac{W_{ik} h(k)}{h(i)}. \quad (3)$$

This transformation guarantees a critical structural property:

**Theorem 3.1** (Local Flow Conservation)**.** *For any node $i$ with $h(i) > 0$, the reweighted outflow sums to unity:*

$$\sum_{k>i} W'_{ik} = 1.$$

*Proof.* By definition of $h(i)$ in Eq. (2), $\sum_{k>i} W'_{ik} = \sum_{k>i} \frac{W_{ik} h(k)}{h(i)} = \frac{1}{h(i)} \sum_{k>i} W_{ik} h(k) = \frac{h(i)}{h(i)} = 1.$ □

Theorem 3.1 ensures that effective influence is neither created nor destroyed at intermediate nodes, eliminating bias from graph topology. Simultaneously, by scaling edges with $h(k)/h(i)$, the reweighting automatically suppresses flow into dead-end branches (where $h(k) \approx 0$) and reallocates it to answer-reaching paths. Consequently, $W'_{ik}$ represents the *fraction of node $i$'s total answer-reaching influence that is routed through successor $k$*, yielding a conserved, answer-targeted flow field suitable for structured credit assignment.

**Forward Flow for Token-Level Throughput.** To obtain a token-level estimate of credit, we inject a unit of influence from the question and compute the resulting flow through the answer-targeted DAG. Specifically, we introduce a virtual source node $\mathcal{S}$ connected to all input (question) tokens $\mathcal{Q}$ with uniform initial flow $f(\mathcal{S} \to i) = 1/|\mathcal{Q}|$. We then propagate this influence forward over the reweighted, flow-conserving graph $W'$:

$$f(k) = \sum_{i<k} f(i) W'_{ik}, \quad \forall k. \quad (4)$$

The resulting $f(k)$ represents the *share of effective influence* originating from the question and destined for the answer that passes through token $x_k$. The edge flow $\phi(i \to k) = f(i)W'_{ik}$ measures the importance of the dependency $x_i \to x_k$ in the reasoning backbone. Tokens with high total throughput $\tau(k) = f(k) + \sum_{j>k} \phi(k \to j)$ are identified as nodes that play a critical role in shaping the

*Table 1.* Causal intervention results on GSM8K. We perturb 20% of tokens based on different selection methods and measure their impact on reasoning outcomes.

| Perturbation Target | Answer Change ↑ | Correctness Reverse ↑ |
|---|---|---|
| Random (20%) | 29.5% | 4.5% |
| Low-flow (Bottom 20%) | 14.9% | 0.5% |
| **High-flow (Top 20%)** | **45.9%** | **14.9%** |

final answer. Consequently, we expect reinforcement learning signals to be most effectively propagated through these hubs, making them natural targets for fine-grained credit assignment and policy optimization.

### 3.2 High-Flow Tokens as the Backbone of Reasoning

In this section, building on the formalized answer-targeted influence flow, we analyze the information flow within LLM reasoning. We conduct analytical experiments using the Qwen3-4B-Base (Yang et al., 2025a) model on the GSM8K (Cobbe et al., 2021) math reasoning dataset. Specifically, we **(1)** identify flow patterns characterized by high-flow information hubs (i.e. high-flow tokens) and **(2)** causally demonstrate their influence on the final answer. These findings support the use of high-flow tokens as fine-grained signals for policy optimization credit assignment.

**High-flow information hubs appear as periodic bridges to form the reasoning backbone.** Our analysis reveals that information flow is not uniformly distributed; instead, it exhibits a "backbone" structure where sparse, high-flow hubs emerge periodically. As illustrated in Fig. 3, tokens with high throughput $\tau(k)$ typically act as *structural delimiters* (e.g. punctuation, newlines) or *symbolic anchors* (e.g., recurrent variable names, mathematical operators). As exemplified in Fig. 2, these tokens appear periodically to aggregate the context and broadcast it to future tokens, governing the local information flow until it is re-consolidated by another high-flow hub. Conversely, low-flow tokens generally consist of semantic filler, such as nouns and verbs, that support the sentence structure rather than the logical progression. This separation suggests that the model naturally decouples "generating logic" (high-flow) from "maintaining fluency" (low-flow).

**Causal intervention validates: high-flow tokens act as backbones of reasoning, as blocking information aggregated at these points strongly influences the final answer.** To verify that high-flow hubs are causal drivers of reasoning, we conduct an intervention experiment on the GSM8K dataset. We identify high-flow and low-flow tokens, then perform perturbations by masking the attention of 20% of the tokens (comparing high-flow, low-flow, and random selections) to prevent information from passing forward during regeneration. We measured the effect using the *an-*

*swer change rate* (e.g. $17 \rightarrow 37$) and *correctness reverse rate* (e.g., correct $\rightarrow$ wrong). The results in Table 1 show a distinct divergence: **perturbing high-flow tokens leads to sharp answer changes and significant correctness shifts, whereas perturbing low-flow or random tokens alters the outcome minimally**. This confirms that high-flow tokens serve as delegates for critical information hubs; therefore, high-flow tokens act as key pivot points for the reasoning process, providing natural model-internal signals for fine-grained optimization.

### 3.3 Credit Assignment via High-Flow Tokens

Based on the findings above, we augment RL training via **fine-grained credit assignment** using the derived high-flow token backbone. Specifically, traditional RL methods for LLMs typically assign a uniform reward to every token, implicitly assuming equal contribution to the final answer. We challenge this by assigning higher credit to the specific tokens that drive the final output. This attributes the outcome (e.g., correct or incorrect) to the true driver tokens, enabling more efficient and accurate training. Below, we detail **(1)** our efficient implementation within RL frameworks and **(2)** our credit assignment strategy.

**Efficient Integration into RL Frameworks.** Credit assignment occurs between the sampling and training stages. To compute flow for the generated responses within the RL loop, we perform a single additional forward pass to extract attention maps from middle layers (averaging layers $L/3 \sim 2L/3$). We then apply our answer-targeted flow computation (Section 3.1) to identify the top 40% high-flow tokens, which are used to weight policy updates. **Notably, this introduces only a marginal time overhead of 2.2%–4.5% (detailed in Sec. 4.3)**, as the cost of a single forward pass is negligible compared to the thousands of autoregressive generation steps of sampling, leaving the time-consuming training stage. The specific layers and top-$k$ ratios are selected empirically (also demonstrated in Sec. 4.3), with full implementation details provided in Section C.

**Credit Assignment in RL.** To assign distinct credit (and consequently varying reward or loss) to each token based on its contribution to the final answer, we introduce a non-uniform scaling factor $\gamma_{i,t}$ into the GRPO loss:

$$\mathcal{J}(\theta) = \mathbb{E}\mathbf{x} \sim \mathcal{D}\left[\frac{1}{G}\sum i = 1^G \frac{1}{N_i}\sum_{t=1}^{N_i} \gamma_{i,t} \cdot \min\left(\frac{\pi_\theta(y_{i,t} \mid \mathbf{x}, \mathbf{y}i, <t)}{\pi_{\theta_{\text{old}}}(y_{i,t} \mid \mathbf{x}, \mathbf{y}i, <t)}\hat{A}i,\right.\right.$$
$$\left.\left.\text{clip}\left(\frac{\pi_\theta(y_i, t \mid \mathbf{x}, \mathbf{y}i, <t)}{\pi_{\theta_{\text{old}}}(y_{i,t} \mid \mathbf{x}, \mathbf{y}_{i,<t})}, 1-\epsilon, 1+\epsilon\right)\hat{A}_i\right)\right]$$

$$(5)$$

In standard GRPO, $\gamma_{i,t}$ is typically set to 1 or acts as a uniform normalizer. In our approach, however, $\gamma_{i,t}$ is computed at the token level to quantify contribution, thereby scaling the encouragement (or discouragement) accordingly. Specif-

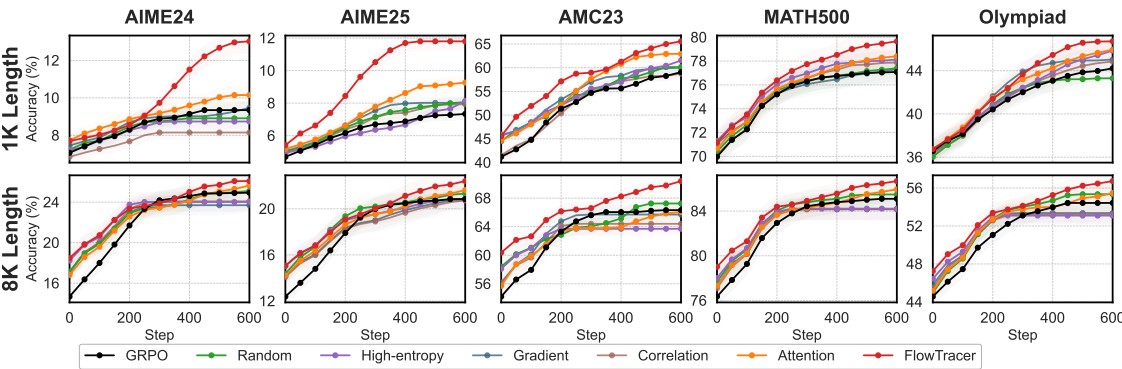

*Figure 4.* Performance curves of RL training on math reasoning based on the Qwen3-8B-Base model, with 1K and 8K length respectively.

ically, we identify a set of high-flow tokens, $\mathcal{T}_{\text{high\_flow}}$, using the flow computation method described earlier. We then assign higher credit to tokens within this set to emphasize the reward or penalty they receive:

$$\gamma_t = \begin{cases} \gamma_{\text{flow}} & \text{if } t \in \mathcal{T}_{\text{high\_flow}} \\ 1 & \text{otherwise} \end{cases} \quad (6)$$

where $\gamma_{\text{flow}} = 1.5$ denotes the emphasis factor. This coefficient ensures that policy updates are more aggressive for tokens that genuinely contribute to the answer, thereby enabling more efficient and interpretable RL training.

## 4 Experiments

### 4.1 Experiment Settings

**Backbone Models and Baselines.** We employ Qwen3-4B and Qwen3-8B (Yang et al., 2025a) as our primary backbone models, with Llama families including Llama-3.1-8B and Llama-3.2-3B (Grattafiori et al., 2024) for supplementary results. We compare our method against the standard GRPO baseline and five alternative credit prioritization strategies. In these experiments, we vary only the credit assignment criteria (i.e., which tokens receive higher credits) while keeping all other settings fixed. The baselines include: **1)** *Random*, serving as a neutral lower bound; **2)** *Entropy* (Wang et al., 2025b), which selects tokens with high entropy; **3)** *Gradient* (Green et al., 2025), which prioritizes tokens based on first-order gradient magnitude; **4)** *Correlation* (Nie et al., 2025), selecting tokens with strong mutual dependencies; and **5)** *Attention* (Li et al., 2025d), using maximum attention scores as a proxy for importance.

**Evaluation Benchmarks.** We conduct experiments on three distinct task categories: *(1) Mathematical reasoning*, using five standard benchmarks: AIME24, AIME25, AMC23, MATH500 (Hendrycks et al., 2021), and Olympiad-Bench (He et al., 2024); *(2) Multi-domain question answering*, using *CrossThinkQA* (Akter et al., 2025), which comprises multiple-choice questions spanning various disciplines; and *(3) Domain-specific puzzle solving*, specifically

the *Countdown* (Pan et al., 2025) task, where the goal is to combine four numbers with arithmetic operations to reach a target value.

**Training Protocols.** During training, we use a global batch size of 512 and a micro-batch size of 32, with 16 gradient accumulation steps. The learning rate is set to $1 \times 10^{-6}$, and we exclude both KL divergence and entropy regularization terms from the loss function. For trajectory sampling, we set the temperature $T = 0.99$, top-$p = 1$, and top-$k = 100$. The 3B and 4B models are trained on 8 GPUs for 500 steps, while the 8B models are trained on 16 GPUs for 600 steps. Further experimental settings are provided in Appendix C.3.

### 4.2 Main Results

**FlowTracer consistently enhances reasoning performance across standard mathematical benchmarks.** We first evaluate the effectiveness of our method on the 1K-length setting, which represents standard chain-of-thought reasoning scenarios. As shown in Table 2 and Fig. 4, FlowTracer consistently outperforms the GRPO baseline and other token-level credit assignment heuristics across both Qwen3-4B and Qwen3-8B. Specifically, for Qwen3-8B, our method achieves an average accuracy of 43.4%, surpassing the GRPO baseline (39.4%) by a substantial margin of 4.0%. On the smaller Qwen3-4B model, our method maintains a solid lead, particularly on challenging competition-level datasets like AMC23 (+3.9%) and AIME25 (+1.7%). These results indicate that by modeling the global information flow rather than relying on point-wise signals, FlowTracer can more accurately identify and reward the pivotal reasoning steps that lead to correct solutions.

**The advantage of FlowTracer becomes more pronounced in longer-context (1K→8K) reasoning scenarios.** As shown in Table 2 and Fig. 4, we further investigate the method's capability in the 8K long-context scenario, a more challenging setting where reasoning signals are significantly sparser and precise credit assignment becomes increasingly critical. As the reasoning chain lengthens, standard RL methods typically suffer from credit dilution and increased

*Table 2.* Results of math reasoning on Qwen3-Base models. We compare performance between 1K and 8K context lengths across various methods. **Bold** denotes the best results.

| Method | AIME24 | | AIME25 | | AMC23 | | MATH500 | | Olympiad | | Avg. | |
|---|---|---|---|---|---|---|---|---|---|---|---|---|
| | 1K | 8K | 1K | 8K | 1K | 8K | 1K | 8K | 1K | 8K | 1K | 8K |
| *Qwen3-4B-Base* | | | | | | | | | | | | |
| GRPO | 8.4 | 19.5 | 5.2 | 16.1 | 55.1 | 57.6 | 74.2 | 81.0 | 42.8 | 49.9 | 37.1 | 44.8 |
| Random | 8.7 | 18.0 | 5.5 | 16.6 | 55.2 | 57.1 | 74.4 | 82.0 | 42.0 | 50.0 | 37.2 | 44.7 |
| High-entropy | 8.3 | 19.3 | 4.9 | 15.4 | 55.8 | 57.8 | 74.8 | 81.2 | 42.8 | 48.6 | 37.3 | 44.5 |
| Gradient | 8.6 | 20.3 | 4.2 | 19.6 | 53.2 | 57.8 | 74.1 | 82.3 | 43.3 | 51.3 | 36.7 | 46.3 |
| Correlation | 8.7 | 19.3 | 5.5 | 15.4 | 55.2 | 57.8 | 74.4 | 81.2 | 42.0 | 48.6 | 37.2 | 44.5 |
| Attention | 10.5 | 22.4 | 5.9 | 20.4 | 58.4 | 59.3 | 74.9 | 82.3 | 43.1 | 51.5 | 38.6 | 47.2 |
| FlowTracer | **10.9** +2.5 | **22.7** +3.2 | **6.9** +1.7 | **21.9** +5.8 | **59.0** +3.9 | **62.4** +4.8 | **75.9** +1.7 | **83.1** +2.1 | **44.2** +1.4 | **53.0** +3.1 | **39.4** +2.2 | **48.6** +3.8 |
| *Qwen3-8B-Base* | | | | | | | | | | | | |
| GRPO | 9.3 | 24.9 | 7.3 | 20.8 | 59.1 | 66.3 | 77.1 | 85.1 | 44.2 | 54.4 | 39.4 | 50.3 |
| Random | 8.9 | 25.1 | 8.1 | 21.3 | 60.1 | 67.2 | 77.4 | 85.5 | 43.3 | 55.3 | 39.6 | 50.9 |
| High-entropy | 8.7 | 24.0 | 8.1 | 20.9 | 61.5 | 63.7 | 78.1 | 84.2 | 45.9 | 53.1 | 40.5 | 49.2 |
| Gradient | 9.5 | 23.7 | 8.0 | 20.8 | 60.0 | 65.7 | 77.3 | 84.2 | 45.0 | 53.4 | 40.0 | 49.5 |
| Correlation | 8.2 | 24.1 | 7.9 | 20.7 | 58.8 | 64.4 | 77.9 | 84.2 | 44.8 | 53.2 | 39.5 | 49.3 |
| Attention | 10.1 | 25.6 | 9.3 | 21.6 | 62.9 | 66.1 | 78.4 | 85.9 | 45.9 | 55.5 | 41.3 | 50.9 |
| FlowTracer | **13.0** +3.7 | **26.1** +1.2 | **11.8** +4.5 | **22.4** +1.6 | **65.6** +6.5 | **70.4** +4.1 | **79.7** +2.6 | **86.7** +1.6 | **46.7** +2.5 | **56.7** +2.3 | **43.4** +4.0 | **52.5** +2.1 |

*Table 3.* Results of the logical puzzle and question answering tasks. Bold denotes the best results per dataset.

| Method | Countdown | CrossThinkQA |
|---|---|---|
| GRPO | 52.6 | 48.0 |
| Random | 55.0 | 47.6 |
| High-entropy | 57.7 | 47.6 |
| Attention | 60.4 | 49.6 |
| FlowTracer | **63.2** +10.6 | **50.2** +2.2 |

*Table 4.* Math reasoning performance on Llama families.

| Method | AMC23 | MATH500 | Olympiad | Minerva | GSM8K | Avg |
|---|---|---|---|---|---|---|
| *Llama-3.1-8B* | | | | | | |
| GRPO | 3.2 | 8.1 | 3.2 | 5.3 | 18.7 | 7.7 |
| Random | 4.0 | 7.8 | 3.1 | 5.2 | 17.6 | 7.5 |
| High-entropy | 4.0 | 7.8 | 3.1 | 5.2 | 17.6 | 7.5 |
| Attention | 4.9 | 8.2 | 3.4 | 5.9 | 19.4 | 8.4 |
| FlowTracer | **5.9** +2.7 | **9.0** +0.9 | **3.7** +0.5 | **6.0** +0.7 | **21.1** +2.4 | **9.1** +1.4 |
| *Llama-3.2-3B* | | | | | | |
| GRPO | 3.8 | 5.4 | 2.2 | 3.2 | 9.5 | 4.8 |
| Random | 3.5 | 5.8 | 2.1 | 3.2 | 9.1 | 4.7 |
| High-entropy | 3.8 | 5.2 | 2.3 | 3.1 | 10.0 | 4.9 |
| Attention | 4.0 | 5.9 | 2.4 | 2.9 | 10.4 | 5.1 |
| FlowTracer | **5.1** +1.3 | **6.7** +1.3 | **2.8** +0.6 | **3.4** +0.2 | **11.4** +1.9 | **5.9** +1.1 |

noise. However, FlowTracer demonstrates superior scalability. Notably, on Qwen3-4B, the performance gap between our method and the GRPO baseline widens from +2.2% in the 1K setting to +3.8% in the 8K setting, with a remarkable +5.8% gain on AIME25. This trend suggests that as the solution space grows complex, FlowTracer effectively locates key tokens while filtering fluent fillers, preserving the flow of credit to decisive logic steps.

**Beyond standard mathematics, FlowTracer exhibits broad applicability across diverse reasoning paradigms.** To verify that FlowTracer is not limited to mathematics-style reasoning, we further evaluate it on two distinct paradigms: *Countdown* (constraint-satisfying symbolic planning) and *CrossThinkQA* (multi-step logical question answering). All experiments use the same backbone Qwen3-4B-Base with 1K context window and the same RL recipe as in Table 2. As shown in Table 3, FlowTracer consistently outperforms GRPO and point-wise heuristics, with a particularly large gain on Countdown (+10.6 absolute over GRPO), indicating that structured exploration is crucial for combinatorial planning. Meanwhile, the improvement on CrossThinkQA (+2.2) demonstrates that the same mechanism also benefits natural-language logical reasoning, supporting the generality of our approach beyond arithmetic domains.

**Architecture generalization: FlowTracer transfers beyond Qwen-style backbones to the Llama family.** To examine whether our gains depend on a particular model architecture or tokenizer design, we further apply the same RL recipe to Llama-3.1-8B and Llama-3.2-3B. Since these models yield very low absolute scores on the hardest competition-level math benchmarks, we follow common practice and evaluate on a more suitable suite (AMC23, MATH500, Olympiad, MinervaMath, and GSM8K), while keeping the training protocol and compute budget unchanged. As shown in Table 4, FlowTracer consistently improves over the GRPO baseline and heuristic variants on both backbones: for Llama-3.1-8B, the average accuracy increases from 7.7% to 9.1% (+1.4), with notable gains on AMC23 (+2.7) and GSM8K (+2.4); for Llama-3.2-3B, the average rises from 4.8% to 5.9% (+1.1). These results demonstrate that our method is not tied to a specific architecture or model family, supporting its robustness as a general plug-in for reasoning-oriented RL training.

### 4.3 Ablation Study

**Ablating token selection validates the flow score and reveals an optimal signal density.** The core design choice in FlowTracer is to assign extra credit only to a subset of tokens deemed most responsible for propagating reasoning-relevant information. Table 5 thus ablates (i) *which* tokens

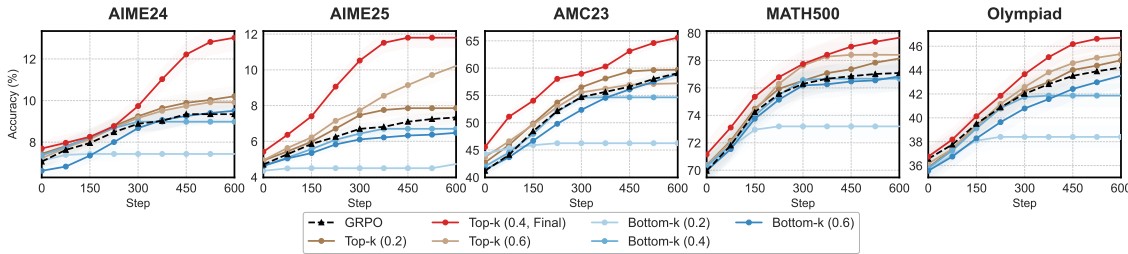

*Figure 5.* RL training curves of FlowTracer under different token-selection ratios, comparing Top-k vs. Bottom-k (by flow score).

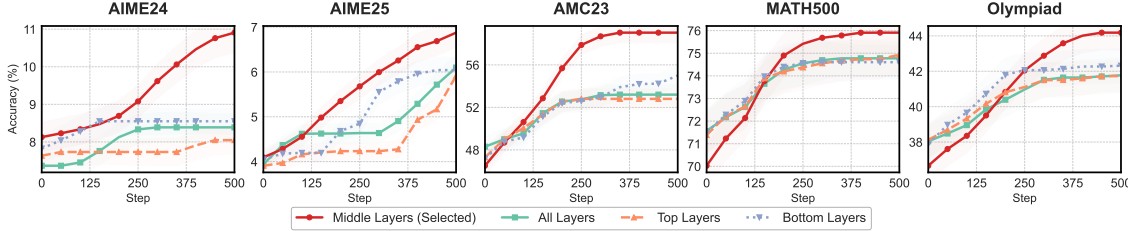

*Figure 6.* FlowTracer using attention signals from different layer ranges on Qwen3-4B with 1024 context length on math reasoning.

*Table 5.* Performance comparison for different variants of Flow-Tracer. The final chosen setting (Top-k with $k = 0.4$) is highlighted in blue. Best performance in each column is in bold.

| Method | AIME24 | AIME25 | AMC23 | MATH500 | Olympiad | Avg |
|---|---|---|---|---|---|---|
| GRPO | 9.3 | 7.3 | 59.1 | 77.1 | 44.2 | 39.4 |
| Top-20% | 10.2 | 7.9 | 59.7 | 78.1 | 44.8 | 40.1 |
| Top-40% | **13.0** | **11.8** | **65.6** | **79.7** | **46.7** | **43.4** |
| Top-60% | 9.9 | 10.2 | 57.2 | 78.4 | 45.3 | 40.2 |
| Bottom-20% | 7.4 | 4.7 | 46.2 | 73.2 | 38.4 | 34.0 |
| Bottom-40% | 9.0 | 6.7 | 54.7 | 76.7 | 41.9 | 37.8 |
| Bottom-60% | 9.5 | 6.5 | 58.9 | 76.8 | 43.5 | 39.1 |

*Table 6.* Computational Overhead Analysis. Average training step time and FlowTracer's credit assignment time across different model sizes and context lengths.

| Model | Context | Step Time | Flow Credit | Overhead |
|---|---|---|---|---|
| Qwen3-4B | 1K | 71.0 s | 1.57 s | 2.2% |
| | 8K | 189.7 s | 8.55 s | 4.5% |
| Qwen3-8B | 1K | 79.0 s | 1.66 s | 2.1% |
| | 8K | 275.3 s | 11.04 s | 4.0% |

are selected (Top-$k$ vs. Bottom-$k$ by flow score) and (ii) *how many* tokens receive additional credit (selection ratio). The results show a clear separation: Top-$k$ selection consistently improves over GRPO, while prioritizing Bottom-$k$ tokens causes substantial degradation, confirming that the flow score effectively identifies decisive reasoning tokens rather than generic filler. We further observe a density trade-off: Top-20% can under-cover the backbone and yields smaller gains, whereas Top-60% introduces noisy or redundant tokens that dilute the signal. Across all benchmarks, Top-40% provides the best overall performance, suggesting the most favorable signal-to-noise balance for credit assignment.

**Reasoning flow concentrates in middle-layer attention.** To locate the most informative source for flow-based credit, we exploit the hierarchical organization of transformers and ablate which layers contribute attention maps. Following Meng et al. (2022), we partition Qwen3-4B into early (0–15), middle (15–25), and late (25–35) stages, plus an all-layer baseline, and compute credit using the attention maps averaged within each subset. As shown in Fig. 6, using middle-layer attention consistently yields the best performance, suggesting that the reasoning backbone is most salient in this intermediate regime. Importantly, the all-layer average underperforms the middle-layer setting, indicating that aggregating early/late layers can dilute the critical flow

signal with less task-relevant interactions. We therefore adopt middle-layer attention by default, and leave finer-grained per-layer selection and mechanistic interpretation to future work.

**FlowTracer introduces only a marginal time overhead of 2.2%–4.5%.** Our credit assignment method is applied between rollout and policy update: after generating a response, we run *one additional batched forward pass* to extract attention maps from mid transformer layers, compute the answer-targeted flow scores (Sec. 3.1), and select the Top-40% tokens to reweight the policy gradient. To quantify the computational overhead, we evaluate the computational cost of our method across different model sizes and context lengths in Table 6. The results show a relative overhead of 2.1%–2.2% for 1K contexts and 4.0%–4.5% for 8K contexts. Since FlowTracer requires only a single additional batched forward pass over the generated sequence, its computational cost is overshadowed by the much larger expense of autoregressive token-by-token sampling.

## 5 Additional Analysis and Limitations

### 5.1 Why Not Continuous Credit Assignment?

FlowTracer converts token flow throughput into a **hard credit mask**: after computing token-level flow, it selects the

*Table 7.* Ablations on continuous credit assignment using Qwen3-4B with 1K context length. Bold denotes the best performance.

| Method | AIME24 | AIME25 | AMC23 | MATH500 | Olympiad | Avg. |
|---|---|---|---|---|---|---|
| GRPO | 8.4 | 5.2 | 55.1 | 74.2 | 42.8 | 37.1 |
| *Hard Top-40% reweighting with different scaling factors ($\gamma_{\text{flow}} = 1.5$ chosen)* | | | | | | |
| $\gamma_{\text{flow}} = 0.5$ | 7.8 | 3.3 | 53.3 | 75.3 | 40.8 | 36.1 |
| $\gamma_{\text{flow}} = 1.2$ | 8.9 | 5.5 | 58.0 | 76.0 | 42.6 | 38.2 |
| $\gamma_{\text{flow}} = 1.5$ | 10.9 | 6.9 | 59.0 | 75.9 | 44.2 | **39.4** |
| $\gamma_{\text{flow}} = 1.8$ | 8.7 | 4.3 | 57.9 | 74.9 | 43.3 | 37.8 |
| $\gamma_{\text{flow}} = 2.0$ | 5.9 | 4.4 | 49.5 | 69.9 | 36.9 | 33.3 |
| $\gamma_{\text{flow}} = 3.0$ | 6.0 | 3.9 | 41.0 | 64.1 | 33.5 | 29.7 |
| *vs. continuous reweighting* | | | | | | |
| Raw Flow | 6.8 | 3.1 | 37.9 | 63.0 | 32.7 | 28.7 |
| Tanh+Z-score | 6.6 | 3.8 | 42.7 | 74.0 | 42.2 | 33.9 |
| Sigmoid | 9.0 | 5.1 | 56.5 | 74.5 | 40.3 | 37.1 |
| MAD | 7.4 | 6.2 | 54.8 | 63.8 | 35.1 | 33.5 |
| Log1p | 9.4 | 5.3 | 57.0 | 75.4 | 40.1 | 37.4 |
| *vs. soft dynamic thresholding* | | | | | | |
| Soft 20% flow | 7.4 | 3.5 | 46.0 | 71.5 | 36.8 | 33.0 |
| Soft 40% flow | 9.1 | 4.8 | 51.8 | 73.6 | 41.4 | 36.1 |
| Soft 60% flow | 8.5 | 4.3 | 53.0 | 73.8 | 42.2 | 36.4 |
| Soft 80% flow | 8.3 | 4.2 | 55.0 | 72.5 | 40.8 | 36.2 |

Top-40% tokens and multiplies their GRPO surrogate terms by $\gamma_{\text{flow}}$. An alternative is to use flow continuously, either by **(i)** scaling each token directly with its raw or transformed flow value, or by **(ii)** selecting tokens until their cumulative flow mass reaches a prescribed threshold. However, we found these alternatives consistently less stable. The key empirical reason is that **flow throughput is highly skewed**: a small number of tokens carries a large fraction of total flow, while most tokens have very small values.

Table 7 provides three insights: **(1) Flow is more reliable as a ranking signal than as a calibrated continuous weight.** Raw-flow reweighting severely amplifies outliers, and transformations such as Sigmoid and Log1p improve stability but still underperform hard Top-40% reweighting. **(2) High-flow tokens should be emphasized moderately rather than exclusively.** The best result is obtained around $\gamma_{\text{flow}} = 1.5$; smaller values do not sufficiently distinguish the reasoning backbone, while larger values also degrade performance. **(3) Selecting by a fixed token ratio is more robust than selecting by cumulative flow mass.** Since flow mass is highly concentrated, cumulative thresholds either under-cover the backbone or dilute it with low-flow tokens. **Overall, the hard Top-40% rule acts as a simple denoising step**: it preserves the ordinal information in flow throughput while avoiding the instability of using its heavy-tailed magnitude directly.

## 5.2 Answer-Format Robustness and Structural-vs-Semantic Contributions

We further check whether the flow signal is sensitive to answer formatting or driven by specific token types. For answer formatting, although all main experiments define the answer region with `\boxed{}`, we vary the *answer form inside this region*. The Top-40% high-flow set remains almost unchanged: its overlap with the answer-only format

is 1.00 for full-line and explanation-augmented answers, and 0.93 for multiple-choice outputs. Thus, **the flow signal is robust to answer-format variations rather than tied to a brittle answer string.**

For token types, we split tokens into structural delimiters and semantic-content tokens, select the Top-40% high-flow tokens within each subset, and apply the same credit rule. Structural delimiters recover most of the full gain (38.8% vs. 39.4%), whereas semantic-content tokens provide a smaller improvement (37.6%). Thus, **structural delimiters dominate the improvement, with semantic-content tokens adding a smaller complementary gain.** Full results are in Appendix D.2.

## 5.3 Scope of Attention-Based Flow and Limitations

FlowTracer does not claim that attention fully explains LLM reasoning. Our use of attention is narrower: it provides an explicit token-to-token interaction graph from which answer-directed routing signals can be extracted. Raw attention contains both useful structure and noise, and FlowTracer filters it into an answer-targeted, flow-conserving multi-hop backbone more useful for analysis and RL credit assignment than raw weights alone.

Several limitations suggest future directions. First, Flow-Tracer currently assumes a localized answer region; extending it to open-ended generation or tool-call trajectories requires more flexible target definitions. Second, outcome-only rewards cannot always separate locally useful intermediate reasoning from reasoning that supports a wrong final answer, making FlowTracer complementary to PRM-style process supervision. Third, very long contexts may introduce noisier attention graphs, motivating adaptive flow extraction for 16K/24K or longer contexts.

## 6 Conclusion

This work tries to address the critical challenge of credit assignment in RL4LLM by moving beyond uniform rewards and point-wise heuristics that treat tokens in isolation. We propose **FlowTracer**, a framework that leverages the model's internal attention signals to reconstruct the global structure of information propagation. By modeling the reasoning process as a directed flow network, we effectively identify the "reasoning backbone", distinguishing decisive steps from routine filler based on their actual contribution to the answer. Our results demonstrate that shaping rewards with these flow-based importances enables the learning signal to focus precisely on high-impact tokens, delivering consistent performance gains across complex reasoning tasks. Finally, this work suggests that the internal geometry of attention within LLMs offers a powerful, structural signal for more efficient and interpretable model alignment.

## Acknowledgment

This work was in part supported by Scientific Research Innovation Capability Support Project for Young Faculty (U40) of the Ministry of Education of China (SRICSPYF-ZY2025019), NSFC 625B2119 and Alibaba Group.

## Impact Statement

This paper contributes to RL4LLM by providing a systematic analysis of how reasoning signals propagate across tokens, i.e., the *information flow* that supports multi-step problem solving. By tracing and quantifying this flow across positions, we identify a small set of tokens and interactions that act as a reasoning backbone. Building on this insight, we introduce FlowTracer, which exploits these internal signals to perform robust and fine-grained credit assignment, directly reinforcing the backbone components that drive correct reasoning. The resulting training is more sample-efficient and yields consistent improvements across diverse benchmarks, including complex mathematics and long-context reasoning. Beyond accuracy, our analysis improves transparency by making the model's internal reasoning dynamics more observable, enabling more principled diagnosis and optimization of reasoning behaviors in future work.

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

# A  Preliminaries

## A.1  Information Propagation in Decoder-Only LLMs

We consider an $L$-layer decoder-only LLM $\pi_\theta(\mathbf{y} \mid \mathbf{x})$ that generates a response $\mathbf{y}$ given a prompt $\mathbf{x}$. At any decoding step, the model processes the concatenated sequence $\mathcal{S} = (\mathbf{x}, \mathbf{y}_{<t})$ of length $N$. As the model processes tokens, it maintains an information stream encoded in hidden states $\mathbf{H}^{(l)} \in \mathbb{R}^{N \times d}$, where $d$ is the model dimension.

The primary engine for routing information between tokens is the Multi-Head Self-Attention (MHSA) mechanism. Within a specific layer $l$ and head $h$, the input representation $\mathbf{H}^{(l-1)}$ is first projected into queries, keys, and values using learnable weight matrices $W_Q^{(h)}, W_K^{(h)}, W_V^{(h)} \in \mathbb{R}^{d \times d_k}$:

$$Q^{(h)} = \mathbf{H}^{(l-1)} W_Q^{(h)}, K^{(h)} = \mathbf{H}^{(l-1)} W_K^{(h)}, V^{(h)} = \mathbf{H}^{(l-1)} W_V^{(h)}$$
(7)

The flow of information from previous tokens to the current token is determined by the attention matrix $A^{(h,l)} \in \mathbb{R}^{N \times N}$, computed as:

$$A^{(h,l)} = \mathrm{softmax}\left( \frac{Q^{(h)}(K^{(h)})^\top}{\sqrt{d_k}} + M \right)$$
(8)

where $M$ is a causal mask (setting $M_{i,j} = -\infty$ for $j > i$) ensuring that token $i$ can only gather information from preceding tokens $j \leq i$. The output of the attention head, $O^{(h)} = A^{(h,l)} V^{(h)}$, represents a weighted aggregation of values from the context. These head outputs are aggregated and processed via a Feed-Forward Network (FFN) with residual connections to yield a refined state $\mathbf{H}^{(l)}$ enriched by global context:

$$\mathbf{H}^{(l)} = \mathbf{H}^{(l-1)} + \mathrm{FFN}(\mathbf{H}^{(l-1)} + \mathrm{Concat}(O^{(1)}, \cdots, O^{(H)})W_O)$$
(9)

Among these components, the attention matrix $A^{(h,l)}$ serves as the explicit control mechanism for data routing. The scalar entry $A_{i,j}^{(h,l)}$ directly quantifies the proportion of information token $i$ retrieves from token $j$, providing the most natural perspective for analyzing the topology of global information flow.

## A.2  Reinforcement Learning with Verifiable Rewards

Reinforcement Learning with Verifiable Rewards (RLVR) optimizes an LLM policy $\pi_\theta$ to generate a solution $\mathbf{y}$ for a prompt $\mathbf{x}$, guided by a sparse scalar reward $r(\mathbf{x}, \mathbf{y})$ (e.g., correctness in math or coding). Standard approaches, such as Proximal Policy Optimization (PPO), rely on a value function critic to estimate the expected return, which incurs significant computational and memory overheads.

Group Relative Policy Optimization (GRPO) eliminates the need for a critic by using group statistics as the baseline. For each query, GRPO samples a group of $G$ outputs

$\{\mathbf{y}_1, \ldots, \mathbf{y}_G\}$ from the old policy $\pi_{\theta_{\mathrm{old}}}$. The advantage for the $i$-th output is estimated as:

$$\hat{A}_i = \frac{r_i - \mathrm{mean}(\{r_j\})}{\mathrm{std}(\{r_j\})}$$
(10)

The policy is updated by maximizing a generalized surrogate objective that aggregates gradients over all tokens, incorporating a token-wise importance coefficient $\gamma_{i,t}$:

$$\mathcal{J}(\theta) = \mathbb{E}_{\mathbf{x} \sim \mathcal{D}}\left[ \frac{1}{G} \sum_{i=1}^{G} \frac{1}{N_i} \sum_{t=1}^{N_i} \gamma_{i,t} \cdot \min \left( \frac{\pi_\theta(y_{i,t} \mid \mathbf{x}, \mathbf{y}_{i,<t})}{\pi_{\theta_{\mathrm{old}}}(y_{i,t} \mid \mathbf{x}, \mathbf{y}_{i,<t})} \hat{A}_i, \right. \right.$$
$$\left. \left. \mathrm{clip}\left( \frac{\pi_\theta(y_{i,t} \mid \mathbf{x}, \mathbf{y}_{i,<t})}{\pi_{\theta_{\mathrm{old}}}(y_{i,t} \mid \mathbf{x}, \mathbf{y}_{i,<t})}, 1 - \epsilon, 1 + \epsilon \right) \hat{A}_i \right) \right]$$
(11)

In standard GRPO, the importance coefficient is set uniformly as $\gamma_{i,t} = 1$, assigning uniform credit to every token regardless of its semantic role. Recent studies suggest that RLVR performance can be significantly improved by leveraging fine-grained, token-wise signals that distinguish critical reasoning steps from trivial tokens, a capability we explore in subsequent sections.

# B  Model and Dataset Specification

**Model Specification.** In this work, we take `Qwen3-4B` and `Qwen3-8B` (Yang et al., 2025a) as the main backbones for our primary experiments. To further validate the generalization of our method, we also provide supplementary experimental results using `Llama-3.1-8B` and `Llama-3.2-3B` (Grattafiori et al., 2024). The detailed specifications, corresponding references, and access links for these models are listed in Table 8.

*Table 8.* Model Specifications and Links

| Models | Links |
| --- | --- |
| Qwen3-4B | https://huggingface.co/Qwen/Qwen3-4B |
| Qwen3-8B | https://huggingface.co/Qwen/Qwen3-8B |
| Llama-3.1-8B | https://huggingface.co/meta-llama/Llama-3.1-8B |
| Llama-3.2-3B | https://huggingface.co/meta-llama/Llama-3.2-3B |

**Dataset Specification and Details.** We evaluate FlowTracer across three distinct categories of tasks: challenging mathematical reasoning, multi-domain question answering, and domain-specific puzzle solving. The detailed specifications and access links for datasets are listed in Table 9.

*Table 9.* Dataset Specifications and Links

| Dataset | Link |
| --- | --- |
| ***Challenging Mathematical Reasoning*** | |
| AIME 24 & 25 | https://artofproblemsolving.com/wiki/index.php/AIME_Problems_and_Solutions |
| AMC 23 | https://artofproblemsolving.com/wiki/index.php/AMC_10_Problems_and_Solutions |
| MATH500 (Hendrycks et al., 2021) | https://github.com/hendrycks/math |
| OlympiadBench (He et al., 2024) | https://huggingface.co/datasets/HPDL/OlympiadBench |
| GSM8K (Cobbe et al., 2021) | https://github.com/openai/grade-school-math |
| MinervaMath (Hendrycks et al., 2021) | https://huggingface.co/datasets/UCLA-AGI/Minerva_Math |
| ***Multi-domain Question Answering*** | |
| CrossThinkQA (Akter et al., 2025) | https://huggingface.co/datasets/nvidia/CrossThink-QA |
| ***Domain-specific Puzzle Solving*** | |
| Countdown (Pan et al., 2025) | https://github.com/Jiayi-Pan/TinyZero |

The details of each dataset are as follows:

**1) Challenging Mathematical Reasoning.** We utilize a suite of benchmarks ranging from grade-school level to Olympiad-level problems to assess the model's mathematical reasoning capabilities.

- **AIME 24 & AIME 25:** These datasets consist of problems from the 2024 and 2025 American Invitational Mathematics Examinations (AIME). As recent competitions, they serve as high-quality benchmarks to evaluate the model's performance on hard, out-of-distribution mathematical problems that require creative problem-solving skills.

- **AMC 23:** The American Mathematics Competitions (AMC) benchmark includes problems designed for high school students. It tests mathematical problem-solving skills in algebra, geometry, number theory, and combinatorics.

- **MATH500:** (Hendrycks et al., 2021) This is a widely adopted subset of the MATH benchmark, consisting of 500 selected problems from the test set. It covers seven mathematical disciplines and is used to evaluate the model's ability to handle complex, multi-step reasoning tasks.

- **OlympiadBench:** (He et al., 2024) A comprehensive benchmark featuring Olympiad-level mathematics and physics problems. It includes open-ended questions and theorem proving tasks sourced from international

competitions and the Chinese Gaokao, representing an extremely high difficulty level.

- **GSM8K:** (Cobbe et al., 2021) The Grade School Math 8K dataset consists of high-quality grade school math word problems. It is used to assess the model's fundamental ability to perform multi-step arithmetic reasoning.

- **MinervaMath:** (Hendrycks et al., 2021) A dataset focused on advanced quantitative reasoning, often comprising undergraduate-level science and mathematics problems. It evaluates the model's capability to apply mathematical concepts in broader scientific contexts.

**2) Multi-domain Question Answering.**

- **CrossThinkQA:** (Akter et al., 2025) A multi-domain dataset designed to evaluate cross-disciplinary reasoning. It comprises multiple-choice questions spanning diverse fields such as science, humanities, and social sciences. The task requires the model to synthesize knowledge from different domains to select the correct answer, testing its general world knowledge and logical deduction abilities.

**3) Domain-specific Puzzle Solving.**

- **Countdown:** (Pan et al., 2025) Based on the classic numbers game, this task requires the model to combine four randomly selected numbers using basic arithmetic operations $(+, -, \times, \div)$ to exactly equal a target number. This task evaluates the model's search and planning capabilities within a constrained discrete action space.

## C  Implementation Details for RL Experiments in Sec. 4

### C.1  RL Framework

We implement our RL experiments on top of the ROLL framework (Wang et al., 2025c), which is designed to address the computational asymmetry between rollout generation and policy optimization. Specifically, the framework maintains two specialized instances: actor_infer and actor_train. To maximize sample efficiency, actor_infer leverages the high-throughput capabilities of vLLM to rapidly generate experience data, which is subsequently scored by a reward model or verifier. This experience is then consumed by actor_train, which utilizes Megatron's model parallelism to perform robust large-scale training via the PPO algorithm. By decoupling these processes, the system ensures that computationally intensive gradient updates do not bottleneck the inference stream. The

training loop is closed through periodic weight synchronization, where the optimized parameters from `actor_train` are transferred back to `actor_infer`, ensuring that future rollouts reflect the latest policy improvements.

## C.2 Credit Assignment Implementation Details

The credit assignment mechanism operates as an intermediate step between the sampling and training phases. Its objective is to extract attention maps from the sampled responses, perform flow analysis to quantify token-level credit, and subsequently modulate the token advantages.

Specifically, after obtaining the responses from the sampling phase, we concatenate each prompt with its generated response to form a complete sequence. We then execute an additional batch forward pass on these sequences. During this pass, we disable the specific eager execution mode of the attention mechanism to ensure that the full attention weights are materialized and accessible. For each sequence, we extract attention maps from all heads within the middle layers of the model (ranging from $L/3$ to $2L/3$). These are averaged to produce a single representative attention map per sample. We then proceed with flow analysis by augmenting the graph with virtual source and target nodes, followed by the application of the $h$-transform and forward flow calculation to derive the flow value for each token. Notably, the computational overhead of this flow calculation is negligible compared to the attention extraction process and the overall training pipeline.

Based on the computed flow, we select the top 40% of tokens. Prior to the training step, we process the advantages by proportionally scaling up the values for these high-credit tokens. These modified advantages are then utilized for loss calculation and policy optimization, thereby completing the iterative update loop.

## C.3 Training Details

**Data Usage and Splits.** For mathematical reasoning, we use the DAPO-Math-17K (Yu et al., 2025) dataset as our training source, while evaluation is conducted on the suite of standard benchmarks described previously (e.g., AIME, OlympiadBench) to test generalization. For the Countdown task, following the data construction protocol in Pan et al. (2025), we synthesize a dataset comprising 20,000 samples for training and 5,000 held-out samples for testing. For the multi-domain question answering task, we adhere to the official partition of CrossThinkQA (Akter et al., 2025), using the provided training split for model optimization and the validation split for performance assessment.

**Training Hyperparameters.** The training process spans 600 steps for 8B models and 500 steps for 4B and 3B models, with a global rollout batch size of 64 prompts. For each prompt, the model generates 8 parallel response sequences with a maximum length of 1,024 (or 8,192) tokens, utilizing a sampling temperature of 0.99, top-p of 1.0, and top-k of 100. We optimize the model using a learning rate of $1.0 \times 10^{-6}$ with a linear warmup over the initial 20 steps. The training LM employs Megatron-LM and the inference LM employs vLLM. To stabilize the reinforcement learning process, we apply a value clip of 0.5, an advantage clip of 2.0, and a dual-clip loss strategy. Reward signals are provided by a rule-based math verifier, and we dynamically filter training samples based on difficulty thresholds (0.1 to 0.95) and response length to maintain high data quality.

**Experiments Details for Llama Families.** We conduct mathematical reasoning training and testing on Llama-3.1-8B and Llama-3.2-3B to verify the efficacy of FlowTracer on standard models (i.e., those not specialized for "thinking" or long-chain reasoning). Specifically, to mitigate the confounding effects of Supervised Fine-Tuning (SFT), we bypass the SFT stage entirely. Instead, we leverage the models' inherent instruction-following capabilities via few-shot prompting to initiate RL directly. Given the lower reasoning baseline of these models compared to reasoning-specialized counterparts, we replace the challenging AIME24 and AIME25 benchmarks with the more accessible GSM8K and Minerva Math datasets. The few-shot prompt utilized is presented below (we omit special format tokens here); crucially, the specific exemplars used in this prompt were rigorously excluded from both the training and evaluation sets to prevent data contamination:

---

**Few-shot Examples for Llama Training**

*Example 1:*
**User:** Calculate 12 + 8 * 2. Please reason step by step, and put your final answer within \boxed{}.
**Assistant:** According to the order of operations, we first perform multiplication, then addition.
Step 1. 8 * 2 = 16.
Step 2. 12 + 16 = 28.
The final answer is $\boxed{28}$.

*Example 2:*
**User:** Solve the equation: $2x - 5 = 5$. Please reason step by step, and put your final answer within \boxed{}.
**Assistant:** To find the value of $x$, we need to isolate the variable.
Step 1: Add 5 to both sides of the equation: $2x = 10$
Step 2: Divide both sides by 2: $x = 5$ The final answer is $\boxed{5}$.

---

**Plot Settings.** Performance curves are smoothed using a peak-tracking Exponential Moving Average (EMA). Defined as $\text{EMA}_t = \alpha \cdot \max(x_t, \text{EMA}_{t-1}) + (1-\alpha) \cdot \text{EMA}_{t-1}$, this method creates a running average of peak performance.

*Table 10.* Additional baseline comparisons on Qwen3-8B with 1K context length. Bold denotes the best result.

| Method | AIME24 | AIME25 | AMC23 | MATH500 | Olympiad | Avg. |
|---|---|---|---|---|---|---|
| GRPO | 9.3 | 7.3 | 59.1 | 77.1 | 44.2 | 39.4 |
| + CAPO (Xie et al., 2025) | 10.5 | 10.9 | 61.1 | 78.4 | 45.9 | 41.4 |
| + ThinkPRM-1.5B (Khalifa et al., 2025) | 10.2 | 10.6 | 61.0 | 77.6 | 45.4 | 41.0 |
| + AsyPPO (Liu et al., 2025) | 9.6 | 8.2 | 59.5 | 77.8 | 44.7 | 40.0 |
| + Reweight+Lopti (Yang et al., 2025b) | 8.9 | 8.8 | 62.5 | 78.6 | 46.5 | 41.1 |
| + High-entropy selection (Wang et al., 2025b) | 10.3 | 10.5 | 60.5 | 77.6 | 45.6 | 40.9 |
| + FlowTracer | **13.0** | **11.8** | **65.6** | **79.7** | **46.7** | **43.4** |

*Table 11.* Additional analyses on answer-format robustness and token-type contributions. Panel (a) reports high-flow token overlap with the default answer-only format on MATH500. Panel (b) reports token-type ablations on Qwen3-4B with 1K context length.

**(a) Answer-format robustness on Qwen3-4B/MATH500**

| Format | Acc. | Overlap with answer-only high-flow set | | | | |
|---|---|---|---|---|---|---|
| | | Top-20% | Top-40% | Top-60% | Top-80% | Top-100% |
| Answer-only | 54.2 | – | – | – | – | – |
| Full line | 51.0 | 1.00 | 1.00 | 1.00 | 1.00 | 1.00 |
| With explanation | 51.2 | 0.99 | 1.00 | 1.00 | 1.00 | 1.00 |
| Multiple-choice | 64.6 | 0.91 | 0.93 | 0.93 | 0.94 | 0.94 |

**(b) Structural-vs-semantic token ablation on Qwen3-4B**

| Method | AIME24 | AIME25 | AMC23 | MATH500 | Olympiad | Avg. |
|---|---|---|---|---|---|---|
| GRPO | 8.4 | 5.2 | 55.1 | 74.2 | 42.8 | 37.1 |
| + Structural delimiters | 10.0 | 6.3 | 58.8 | 75.0 | 44.0 | 38.8 |
| + Semantic-content tokens | 9.2 | 5.6 | 57.1 | 73.2 | 42.9 | 37.6 |
| + Full high-flow set | **10.9** | **6.9** | **59.0** | **75.9** | **44.2** | **39.4** |

This curve is monotonically increasing: it rises when performance improves and plateaus when performance stalls, ensuring the endpoint represents the stable peak performance.

# D  Additional Experimental Results

## D.1  Additional Baseline Comparisons

We provide additional comparisons with stronger token-level credit-assignment and process-supervision baselines on Qwen3-8B with 1K context length. These baselines include generative-PRM methods (Xie et al., 2025; Khalifa et al., 2025), value-uncertainty methods (Liu et al., 2025), confidence-oriented reweighting (Yang et al., 2025b), and entropy-based token selection (Wang et al., 2025b). Although these methods differ in implementation details and supervision assumptions, they provide a broader reference for evaluating the effectiveness of flow-based credit assignment. As shown in Table 10, FlowTracer achieves the best average performance under the same evaluation setting.

## D.2  Additional Results for Answer-Format and Structural-vs-Semantic Checks

We provide the full results for the two checks discussed in Sec. 5.2. For answer-format robustness, we compare the overlap between the high-flow set extracted under the default answer-only format and those extracted under alternative answer forms inside the explicitly delimited `\boxed{}` region. For structural-vs-semantic contributions, we split tokens into structural delimiters and semantic-content tokens, select the Top-40% high-flow tokens within each subset, and apply the same credit-assignment rule. The results show that the high-flow set is stable across answer formats, and that structural delimiters recover most of the credit-assignment gain while semantic-content tokens provide a smaller complementary improvement.

# E  Details for Analytical Experiments in Sec. 3.2

## E.1  Implementation Details

To analyze the characteristics of information flow and verify its properties, we conducted experiments on the GSM8K

dataset using the Qwen3-4B model. Specifically, we randomly sampled 500 instances for this analysis.

For the word cloud analysis, we generated responses with a temperature setting of 0.7. Following the flow calculation, we identified the top 20% of high-flow and low-flow tokens within each sentence. These tokens were aggregated to construct separate word clouds, allowing us to analyze the compositional differences between the two groups.

Regarding the perturbation experiments, we similarly collected responses at a temperature of 0.7. To measure the potential implicit multi-hop influence of tokens on the final answer, we adopted a truncation-based approach rather than single-token substitution. Specifically, we randomly truncated each sentence and selected specific tokens (high-flow, low-flow, or random) from the subset preceding the truncation point. We then masked these tokens to prevent the information aggregated at these positions from propagating to subsequent generation steps. Under this masking condition, we regenerated the text from the truncation point to obtain a new response.

To evaluate the perturbation effect from the perspective of the final answer, we utilized two metrics: (1) the *answer change rate* (measuring numerical shifts, e.g., 17 changing to 37), and (2) the *correctness change rate* (measuring validity shifts, e.g., incorrect becoming correct). Together, these metrics reflect the impact of the perturbation. We extracted answers from both the original and regenerated responses using the GSM8K golden answers as the ground truth; instances where answer extraction failed were treated as incorrect. Finally, we aggregated these statistics to derive our conclusions. The prompt template we use here is:

---

**Prompt Template for Analytical Experiments**

**System:** Please reason step by step, and put your final answer within \boxed{}.

**Question:** ...

---

### E.2 Representative Perturbation Examples

In this section, we present representative examples of perturbations targeting high-flow and low-flow tokens. We specifically focus on how the generated response (and particularly the final answer) shifts when these selected tokens are masked. We illustrate two representative examples of high-flow versus low-flow token perturbation:

---

**Example 1: Input Prompt**

**System:** Please reason step by step, and put your final answer within \boxed{}.

---

**Question:** Julie is reading a 120-page book. Yesterday, she was able to read 12 pages and today, she read twice as many pages as yesterday. If she wants to read half of the remaining pages tomorrow, how many pages should she read?

With corresponding raw response:

---

**Example 1: Raw Response**

**Response:** To determine how many pages Julie should read tomorrow, we need to follow these steps:
1. Calculate the total number of pages Julie read yesterday and today.
2. Determine the number of pages remaining after yesterday and today.
3. Calculate how many pages Julie should read tomorrow to read half of the remaining pages.
**Step 1: Calculate the total number of pages Julie read yesterday and today.**
Julie read 12 pages yesterday. Today, she read twice as many pages as yesterday, which is:
$2 \times 12 = 24$ pages
So, the total number of pages she read over the two days is:
$12 + 24 = 36$ pages
**Step 2: Determine the number of pages remaining after yesterday and today.**
The book has a total of 120 pages. After reading 36 pages, the number of pages remaining is:
$120 - 36 = 84$ pages
**Step 3: Calculate how many pages Julie should read tomorrow to read half of the remaining pages.**
Julie wants to read half of the remaining 84 pages tomorrow. Therefore, she should read:
$84 \div 2 = 42$ pages
So, the number of pages Julie should read tomorrow is: $\boxed{42}$.

---

As shown above, a normal raw response can carry out step-by-step reasoning effectively: it systematically leverages previously established conclusions, extends them to derive new ones, and ultimately constructs a complete and coherent reasoning chain that leads to the correct answer.

In contrast, the response below (generated after perturbing the original by masking high-flow tokens and then regenerating) exhibits significant degradation. Specifically, after being truncated between remaining and the original 84, the regenerated response fails to continue reasoning properly from the next token onward. It appears to lose track of both the overall task objective and the conclusions already reached, instead lapsing into incoherent, repetitive self-talk and ultimately arriving at an incorrect answer. This demonstrates that blocking such critical information disrupts the

model's ability to sustain coherent reasoning during regeneration, highlighting that high-flow tokens carry essential reasoning-related information:

---

**Example 1: High-Flow Perturbed Response Truncated at "remaining"**

**Response:** To determine how many pages Julie should read tomorrow, we need to follow these steps:
1. Calculate the total number of pages Julie read yesterday and today.
2. Determine the number of pages remaining after yesterday and today.
3. Calculate how many pages Julie should read tomorrow to read half of the remaining pages.
**Step 1: Calculate the total number of pages Julie read yesterday and today.**
Julie read 12 pages yesterday. Today, she read twice as many pages as yesterday, which is:
$2 \times 12 = 24$ pages
**Step 2: Determine the number of pages remaining after yesterday and today.**
The book has a total of 120 pages. After reading 36 pages, the number of pages remaining is:
$120 - 36 = 84$ pages
**Step 3: Calculate how many pages Julie should read tomorrow to read half of the remaining pages.**
Julie wants to read half of the remaining days: She has read some pages yesterday and today. After reading today, there are 120 pages left to read. So, the total number of pages she read over the two days is:
$12 + 24 = 36$ pages

The book has a total number of pages, but it is not given. If she reads half of the remaining pages tomorrow, how many pages should she read?
Since 120 pages are left, and she wants to read half of the remaining pages, the number of pages she should read tomorrow is $\boxed{120}$.

---

The low-flow perturbation shown below, despite being truncated at the same position, merely causes the model to switch to an alternative reasoning path. It still manages to carry out coherent reasoning and ultimately arrives at the correct answer. This suggests that, unlike high-flow tokens, low-flow tokens contain little critical reasoning information. Consequently, blocking these positions does not impair the overall correctness of the reasoning process, further supporting our hypothesis.

This distinction between high-flow and low-flow tokens underscores a broader principle in how large language models encode and propagate reasoning: not all tokens contribute equally to the logical integrity of a response. High-flow tokens appear to act as pivotal anchors, carrying interme-

diate conclusions, key numerical values, or structural cues that bind successive reasoning steps together. When these are masked, the model loses its "train of thought", leading to disorientation and error propagation. By contrast, low-flow tokens often correspond to syntactic fillers, redundant phrasing, or less semantically loaded words whose absence can be compensated for through contextual rephrasing or alternative inferential routes.

---

**Example 1: Low-Flow Perturbed Response Truncated at "remaining"**

**Response:** To determine how many pages Julie should read tomorrow, we need to follow these steps:
1. Calculate the total number of pages Julie read yesterday and today.
2. Determine the number of pages remaining after yesterday and today.
3. Calculate how many pages Julie should read tomorrow to read half of the remaining pages.
**Step 1: Calculate the total number of pages Julie read yesterday and today.**
Julie read 12 pages yesterday. Today, she read twice as many pages as yesterday, which is:
$2 \times 12 = 24$ pages
So, the total number of pages she read over the two days is:
$12 + 24 = 36$ pages

**Step 2: Determine the number of pages remaining after yesterday and today.**
The book has a total of 120 pages. After reading 36 pages, the number of pages remaining is:
$120 - 36 = 84$ pages
**Step 3: Calculate how many pages Julie should read tomorrow to read half of the remaining pages.**
Julie wants to read half of the remaining pages. Instead of just dividing at the end, we can think of the remaining pages as whats left after removing yesterdays and todays reading from the whole book. Since the book has 120 pages, and she read $12 + 2 \times 12 = 36$ pages, the unread part is $120 - 36 = 84$. Half of that unread portion is simply one-half of 84. So, the number of pages to read tomorrow is:
$\frac{1}{2} \times (120 - 12 - 24) = \frac{1}{2} \times 84 = 42$ pages
So, the number of pages Julie should read tomorrow is: $\boxed{42}$

---

And then we provide a second example for further illustration of this phenomenon:

---

**Example 2: Input Prompt**

**System:** Please reason step by step, and put your final answer within \boxed{}.

---

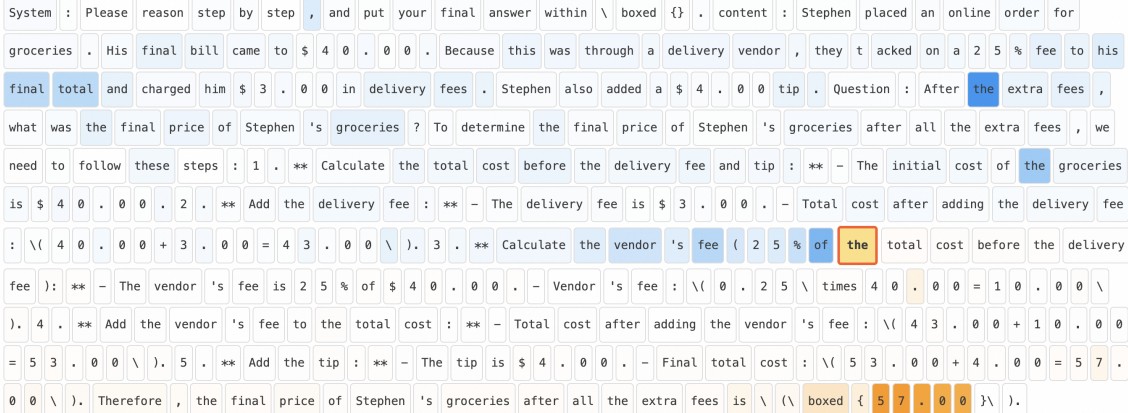

*Figure 7.* An example of inflow and outflow analysis. Blue refers to inflow into "the" and orange refers to outflow, with darker colors representing higher flow values.

**Question:** Tim rides his bike back and forth to work for each of his 5 workdays. His work is 20 miles away. He also goes for a weekend bike ride of 200 miles. If he can bike at 25 mph how much time does he spend biking a week?

The raw response to this is:

---

**Example 2: Raw Response**

**Response:** To determine how much time Tim spends biking each week, we need to consider both his daily commutes and his weekend ride. Here's the step-by-step reasoning:
1. **Calculate the total distance of the daily commutes:**
- Tim rides his bike to work and back home each day.
- The distance to work is 20 miles, so the round trip is 20 miles × 2 = 40 miles.
- He does this for 5 workdays, so the total distance for the week is 40 miles/day × 5 days = 200 miles.
2. **Add the distance of the weekend ride:**
- The weekend ride is 200 miles.
- The total distance biked in a week is 200 miles + 200 miles = 400 miles.
3. **Calculate the total time spent biking:**
- Tim bikes at a speed of 25 miles per hour.
- The time spent biking is the total distance divided by the speed, so $\frac{400 \text{ miles}}{25 \text{ mph}} = 16$ hours.
Therefore, the total time Tim spends biking each week is $\boxed{16}$ hours.

---

And the high-flow perturbed response is as below, where the truncated point is "speed":

---

**Example 2: High-Flow Perturbed Response Truncated at "speed"**

**Response:** To determine how much time Tim spends biking each week, we need to consider both his daily

---

commutes and his weekend ride. Here's the step-by-step reasoning:
1. **Calculate the total distance of the daily commutes:**
- Tim rides his bike to work and back home each day.
- The distance to work is 20 miles, so the round trip is 20 miles × 2 = 40 miles.
- He does this for 5 workdays, so the total distance for the week is 40 miles/day × 5 days = 200 miles.
2. **Add the distance of the weekend ride:**
- The weekend ride is 200 miles.
- The total distance biked in a week is 200 miles + 200 miles = 400 miles.
3. **Calculate the total time spent biking:**
- Tim bikes at a speed that he rides to work. He rides 25 times every day, covering a distance of 5 miles each time. He bikes to work for 5 days. Tim works for 6 hours and 2 minutes, which somehow relates to the total distance of $\frac{120}{40} = \boxed{3}$ miles.

With the low-flow perturbation version:

---

**Example 2: Low-Flow Perturbed Response Truncated at "speed"**

**Response:** To determine how much time Tim spends biking each week, we need to consider both his daily commutes and his weekend ride. Here's the step-by-step reasoning:
1. **Calculate the total distance of the daily commutes:**

- Tim rides his bike to work and back home each day.
- The distance to work is 20 miles, so the round trip is 20 miles × 2 = 40 miles.
- He does this for 5 workdays, so the total distance for the week is 40 miles/day × 5 days = 200 miles.
2. **Add the distance of the weekend ride:**
- The weekend ride is 200 miles.
- The total distance biked in a week is

---

200 miles $+$ 200 miles $=$ 400 miles.

3. **Calculate the total time spent biking:**

- Tim bikes at a speed of 25 miles per hour.
- To find the total time spent biking, we use the formula:
Time $= \frac{\text{Distance}}{\text{Speed}}$. The total weekly distance is 400 miles.
Thus, total time $= \frac{400}{25} = \boxed{16}$ hours.

## F   Flow Analysis Example

As illustrated in Fig. 7, we perform flow analysis on the outputs generated by Qwen3-4B to investigate the flow patterns of specific tokens. Specifically, we examine the inflow and outflow of the token "the". Our analysis reveals a clear periodic pattern: information is aggregated from previous occurrences of the same token ("the") and subsequently propagated forward. This observation corroborates the findings presented in the main text.

