# OpenReview forum: "How Does Reasoning Flow? Tracing Attention-Induced Information Flow for Targeted RL in LLMs"
_ICML.cc/2026/Conference — ICML 2026 regular_

### Official Review · Reviewer_sebK · 2026-02-15

**Soundness:** 3
**Presentation:** 3
**Significance:** 3
**Originality:** 3
**Overall Recommendation:** 5
**Confidence:** 5

**Summary:**

The paper analyzes how reasoning information flows from prompt to answer inside an LLM, and proposes FlowTracer, which traces answer-targeted influence flow on an attention-induced token DAG and scores tokens by flow throughput for token-level credit assignment. Overall, the paper uses these flow-based token importances to shape token-level rewards in GRPO, focusing updates on decisive reasoning hubs and improving performance on reasoning tasks with small overhead.

**Compliance With Llm Reviewing Policy:**

Affirmed.

**Final Justification:**

Good paper! This should be accepted.

**Key Questions For Authors:**

See Weaknesses

**Limitations:**

yes

**Strengths And Weaknesses:**

+ Strengths
  + The paper studies an important research problem that the LLM community cares about: fine-grained (token-level) credit assignment. This is especially useful for long-horizon RL training, where each rollout is expensive and standard GRPO-style updates may not fully exploit the rich information within a trajectory.
  + The proposed method is novel (to my knowledge) and interesting. Using attention-flow signals to identify influential tokens along the prompt→answer information path, and then emphasizing their credit assignment, is intuitive. The overall design also feels relatively elegant.
  + The experimental results look strong, and the empirical execution appears solid.
  + The paper’s explicit analysis of computational cost/overhead is useful.

+ Weaknesses
  + While the method is intuitive, the underlying assumption seems worth scrutinizing. As I understand it, the method identifies “important” tokens and then applies stronger encouragement/penalty depending on whether the final answer is correct. However, in long-horizon settings, it is common that a rollout is globally incorrect while containing locally correct and helpful reasoning steps/tokens that should arguably be reinforced, whereas other later mistakes cause the final failure. It is unclear whether the proposed token-level credit assignment can reliably separate these cases, or whether it may over-penalize useful intermediate reasoning when the final outcome is wrong.
  + I noticed that as context length increases (e.g., 1k → 8k), the improvement over the baseline appears to shrink. Is there a specific discussion or hypothesis for this behavior? Since modern models (e.g., Qwen3) are often used with 16k/24k or longer contexts, it would be helpful to discuss (in words) whether gains may further diminish at larger context lengths, even if running those experiments is computationally expensive.
  + A natural alternative baseline is to use a strong LLM (or other heuristic) to label important tokens and then train a lightweight token-level predictor (e.g., an MLP head) to approximate this importance signal. Even if not evaluated experimentally, it would be useful to discuss how FlowTracer differs from (and potentially improves upon) such “learned token-importance” alternatives, and what trade-offs are expected.

---

> ### Author Rebuttal · Authors · 2026-03-31
>
> We thank the reviewer for the careful reading and encouraging feedback. We highly appreciate the recognition of the importance of fine-grained credit assignment, as well as the reviewer’s positive assessment of the novelty, elegance, empirical strength, and explicit overhead analysis of our method. Below we address the main concerns.
>
> ---
> > **Q1: If a rollout is globally incorrect but contains locally useful reasoning, can the method over-penalize good intermediate steps?**
>
> This is an important question. We agree that, in long-horizon reasoning, a globally incorrect trajectory may still contain locally correct or helpful intermediate steps. FlowTracer is not intended to fully solve this credit-assignment ambiguity in the same way that a perfect process reward model would. Rather, its role is to make sequence-level RL signals more targeted by concentrating updates on the tokens that most strongly route information toward the produced answer.
>
> In this sense, when an answer is wrong, FlowTracer helps the model localize the internal routing backbone that led to that wrong outcome, instead of spreading the same penalty uniformly over all generated tokens. We view this as a refinement over standard GRPO-like RL methods, not as a complete substitute for process-level supervision. The reviewer’s point is well taken: distinguishing “useful intermediate reasoning later followed by a mistake” from “reasoning that directly supports the wrong answer” remains challenging under outcome-only rewards. We will explicitly discuss this limitation in the revision and note that FlowTracer is complementary to PRM-style supervision rather than competing with it.
>
> ---
> > **Q2: The gain appears to shrink as context length increases; what might happen at even longer contexts such as 16K?**
>
> Thank you for highlighting this trend. We agree that long-context behavior deserves a clearer discussion. The current results suggest that the picture is nuanced rather than uniformly diminishing. On **Qwen3-4B**, the advantage over GRPO actually becomes larger in the 8K setting than in the 1K setting, indicating that precise credit assignment can become even more valuable as reasoning length grows. On **Qwen3-8B**, the gain remains positive but is somewhat smaller at 8K, suggesting that the benefit may depend on backbone scale and how well the model already handles long-range dependencies.
>
> Our current hypothesis is that longer contexts create two opposing effects: they increase the need for structured credit assignment because reward dilution becomes more severe, but they also make the attention graph noisier and the answer-reaching backbone harder to isolate cleanly. The net result can therefore vary by model. We will add this discussion in the revision, and if space permits, we will include additional longer-context evidence or preliminary 16K observations.
>
> ---
> > **Q3: How does FlowTracer differ from learned token-importance alternatives, such as training a token-level rewarder or predictor?**
>
> This is a very helpful comparison. We view FlowTracer and learned token-importance methods as related, but importantly different. Learned token-importance approaches rely on an external scorer or predictor that must itself be trained, which introduces additional supervision requirements, model complexity, and potential generalization issues as the policy evolves during RL. In contrast, FlowTracer extracts its signal directly from the current policy’s own internal attention structure, requiring no auxiliary model, no extra labels, and only one additional forward pass.
>
> The trade-off is that a learned importance model may, in principle, capture richer supervisory signals if trained successfully, whereas our method is constrained by the quality of the model-internal proxy. At the same time, external token-importance models can become stale as the policy changes, while FlowTracer is recomputed online from the current policy and therefore naturally adapts throughout training. More broadly, accurately estimating process-level rewards or values remains itself a challenging problem, and existing evidence suggests that critic learning in this setting is nontrivial. We will add this discussion to better position FlowTracer as a lightweight, model-internal alternative that is complementary to external reward-model approaches rather than a replacement for them.
>
> ---
>
> We hope this response clarifies the intended scope of FlowTracer and its relation to longer-horizon reasoning and learned token-importance alternatives. We are grateful for the reviewer’s insightful questions, which will help strengthen the final version of the paper.

---

> > ### Author Rebuttal · Reviewer_sebK · 2026-03-31
> >
> > Thanks for the rebuttal. I think this paper should be accepted and I would like to keep my high score of 5.

---

> > > ### Author Response · Authors · 2026-04-07
> > >
> > > Thank you for your support and for acknowledging our rebuttal. We sincerely appreciate your time, and we would be more than happy to address any remaining questions you might have.

---

### Official Review · Reviewer_gqWU · 2026-03-08

**Soundness:** 3
**Presentation:** 3
**Significance:** 3
**Originality:** 3
**Overall Recommendation:** 5
**Confidence:** 3

**Summary:**

The paper proposes FlowTracer, a token-level credit assignment framework for RL in LLMs that models reasoning as answer-targeted information flow over an attention-induced DAG. A Doob-h-like reweighting enforces local flow conservation and filters paths that do not reach the answer, and token throughput on the resulting flow backbone is used to prioritize credit in GRPO updates. Experiments on math reasoning, long-context settings, a puzzle task, and multi-domain QA show consistent improvements over GRPO and several token-level heuristics, with causal interventions indicating that high-flow tokens are more influential to final answers.

**Compliance With Llm Reviewing Policy:**

Affirmed.

**Final Justification:**

The rebuttal thoroughly addressed my concerns regarding answer region robustness, the role of structural vs. semantic tokens, and the choice of top‑40% threshold. The additional ablation on continuous weighting and the commitment to a limitations section further strengthen the paper. My original score of 5 (Accept) remains unchanged.

**Key Questions For Authors:**

1. What precisely defines the “answer region” across all evaluated tasks, and how sensitive are results to this choice?
2. To what extent are gains driven by structural delimiter tokens (newlines/punctuation) versus semantic content tokens?
3. Why is Top-40% the right operationalization of “throughput credit,” and does continuous weighting help?

**Limitations:**

The paper does not sufficiently discuss the core methodological limitations, especially the reliance on attention as a proxy for causal influence and the potential confound that high-flow tokens are often delimiters/formatting anchors, and the sensitivity to how the “answer region” is defined, which can materially change the flow signal. It should also more explicitly note that improving RL credit assignment can accelerate capability gains and could be misused without appropriate safety constraints, and briefly outline mitigations (e.g., safety-aligned rewards, constrained policies, safety evaluations).

**Strengths And Weaknesses:**

Strengths:
1. The paper pinpoints a concrete failure mode in RL-style fine-tuning for long reasoning: sparse end rewards get “smeared” across tokens, so decisive steps are under-trained.
2.  Modeling a generated response as an attention-induced DAG is intuitive, and it gives the reader a tangible object to reason about rather than vague attribution.
3. The method plugs into GRPO by reweighting per-token loss on a selected subset without modifying the reward definition, sampling procedure, or adding auxiliary models, making it easy to combine with existing RLVR recipes.
4.  The added overhead is cleanly tied to one extra forward pass for attention extraction, which is predictable across model sizes and context lengths and avoids hidden costs from external scorers or complex multi-model pipelines.

Weaknesses：
1. The whole construction is fragile to how you define the answer span. Since reachability and the Doob-h conditioning depend on the “answer region,” small implementation choices (last digit vs full line, multiple-choice letter vs explanation) can change which tokens get emphasized. There isn’t a convincing sensitivity sweep showing the gains are robust.
2. It leans heavily on attention-as-influence. Even with the flow machinery, you’re still building capacities from attention weights; attention is not a reliable causal explanation by default. The current evidence doesn’t rule out that flow is mostly tracking decoding dynamics rather than “information that supports correctness.”
3. The paper doesn’t convincingly show that the method remains useful when the “answer” is not a clean localized substring (open-ended generation, tool calls, multi-field outputs). Right now the evaluation setting is unusually friendly to answer-conditioned credit assignment.

---

> ### Author Rebuttal · Authors · 2026-03-31
>
> We thank the reviewer for the careful reading and encouraging feedback. We especially appreciate the recognition that the attention-induced DAG provides a concrete object for token-level credit assignment and integrates cleanly into GRPO. Below we address the main concerns.
>
> ---
> > **Q1: The method may be fragile to how the answer region is defined.**
>
> Thank you for this point. In our experiments, we define the answer region using an explicitly boxed region (i.e., `\boxed{}`). This provides a stable answer region signal and minimizes ambiguity.
>
> To further verify robustness, we test **whether the high-flow token subset changes when the content inside the boxed region varies**. Using Qwen3-4B on MATH-500, we compare four answer formats (driven by prompts): **(1)** answer only (baseline), **(2)** full line with variables/units, **(3)** answer with a brief explanation, and **(4)** multiple-choice.
>
> As shown in the table below, our flow analysis reveals extremely high consistency across formats. This confirms that our method is robust to the specific content within the answer region and captures intrinsic reasoning checkpoints.
>
> |Format Type|Answer Accuracy|top-20%|top-40%|top-60%|top-80%|top-100%|
> |-|-|-|-|-|-|-|
> |Answer only (Baseline)|54.2%|-|-|-|-|-|
> |Full line (vs Baseline)|51.0%|1.00|1.00|1.00|1.00|1.00|
> |With explanation (vs Baseline)|51.2%|0.99|1.00|1.00|1.00|1.00|
> |Multiple-choice (vs Baseline)|64.6%|0.91|0.93|0.93|0.94|0.94|
>
> ---
> > **Q2: Flow may reflect decoding dynamics rather than information supporting correctness.**
>
> We agree that attention-derived flow should not be framed as a strict causal proof of correctness. Our claim is narrower: it is a useful internal proxy for positions that are structurally important for answer-directed generation, and this proxy is helpful for both analysis and RL credit assignment. We will revise the wording to avoid overclaiming.
>
> ---
> > **Q3: The evaluation is friendly to localized answer substrings; what about open-ended generation or tool calls?**
>
> This is a very good suggestion. We focus on localized answers to ensure tractable analysis and reliable reward signals via clearly defined answer regions. Extending this framework to open-ended generation or tool calls is an important next step, and we will explicitly add this as a limitation and future direction in the revision.
>
> ---
> > **Q4: To what extent are gains driven by structural delimiter tokens (newlines/punctuation) versus semantic content tokens?**
>
> We appreciate this insightful question. To investigate this, we conduct an RL ablation study separating structural delimiter tokens and semantic-content tokens within the high-flow set (Qwen3-4B, 1K context, top 40% selected per subset).
>
> As shown below, **the improvements are primarily driven by structural tokens**. Semantic tokens provide a smaller, supplementary boost. This indicates that **structural tokens form the foundation of the performance boost, while semantic-content tokens serve to further reinforce it**.
>
> |Method|AIME24|AIME25|AMC23|MATH500|Olympiad|Avg|
> |-|-|-|-|-|-|-|
> |GRPO|8.4|5.2|55.1|74.2|42.8|37.1|
> |+ FlowTracer with **structural delimiter tokens**|10.0|6.3|58.8|75.0|44.0|***38.8 (+1.7)***|
> |+ FlowTracer with **semantic-content tokens**|9.2|5.6|57.1|73.2|42.9|37.6 (+0.5)|
> |+ FlowTracer with **full tokens**|10.9|6.9|59.0|75.9|44.2|**39.4 (+2.3)**|
>
> ---
> > **Q5: Why is Top-40% the right choice, and does continuous weighting help?**
>
> The **Top-40%** choice is supported by the ablation in Table 5: smaller subsets under-cover the reasoning steps, while larger ones introduce noise.
>
> We also explored continuous weighting but found it less effective due to the heavy-tailed flow distribution. Takeaways are: **(1)** Raw continuous flow amplifies outliers and severely degrades performance. **(2)** While squashing transformations (e.g., Sigmoid, Log1p) stabilize training back to the baseline, they still underperform the Top-40% approach. **(3) The discrete threshold serves as a necessary and highly effective denoising step**.
>
> |Method|AIME24|AIME25|AMC23|MATH|Olympid|Avg|
> |-|-|-|-|-|-|-|
> |**GRPO**|8.4|5.2|55.1|74.2|42.8|37.1|
> |+ FlowTracer (top-k=40%)|10.9|6.9|59.0|75.9|44.2|**39.4**|
> |+ Continuous **(Raw Flow)**|6.8|3.1|37.9|63.0|32.7|28.7|
> |+ Continuous **(Flow + Tanh Z-score)**|6.6|3.8|42.7|74.0|42.2|33.9|
> |+ Continuous **(Sigmoid)**|9.0|5.1|56.5|74.5|40.3|37.1|
> |+ Continuous **(MAD)**|7.4|6.2|54.8|63.8|35.1|33.5|
> |+ Continuous **(Log1p)**|9.4|5.3|57.0|75.4|40.1|37.4|
>
> ---
> > **Q6: The paper needs a stronger limitations discussion.**
>
> We agree and appreciate the reviewer’s clear summary of the relevant limitations. In the revision, we will add a dedicated limitations section covering the discussions.
>
> ---
> We hope this response clarifies both the current scope and the limitations of our method. We are grateful for the reviewer’s constructive suggestions, which will help us improve the paper’s responsible framing.

---

> > ### Author Rebuttal · Reviewer_gqWU · 2026-04-03
> >
> > We thank the reviewer for the clarification.

---

> > > ### Author Response · Authors · 2026-04-07
> > >
> > > Thank you for your constructive feedback and for confirming that our clarifications have resolved your concerns. Please feel free to let us know if any further questions arise.

---

### Official Review · Reviewer_vfJU · 2026-03-11

**Soundness:** 2
**Presentation:** 3
**Significance:** 1
**Originality:** 3
**Overall Recommendation:** 3
**Confidence:** 4

**Summary:**

This paper proposed a fine-grained-reward-based Reinforcement Learning Method, named FlowTracer. FlowTracer leverages the attention map to estimate the tokens' importance on the reasoning trace. Several experiments preliminarily demonstrate the potential of this method, but convincing experiments are still lacking.

**Compliance With Llm Reviewing Policy:**

Affirmed.

**Final Justification:**

I will maintain my score and preference.

**Key Questions For Authors:**

See Strengths and Weaknesses.

**Limitations:**

yes

**Strengths And Weaknesses:**

##### Strength
- FlowTracer brings a novel and reasonable perspective on modeling token reward.
- The evaluation setting is convincing as several widely-used benchmarks are considered.

##### Weakness
- Weak Baselines: Only one open-sourced baseline (GRPO) is selected, while other baselines are handmade by the authors. More related and open-sourced baselines should be considered.
- Marginal Improvement: Although the idea quitely make sense, the improvement of FlowTracer is negligible. It is even unclear whether the marginal improvement is caused by random fluctuations.

---

> ### Author Rebuttal · Authors · 2026-03-31
>
> We thank the reviewer for the concise assessment and for recognizing that FlowTracer offers a novel and reasonable perspective on fine-grained token reward modeling, as well as for acknowledging the breadth of the evaluation setting. Below we address the two main concerns.
>
> ---
> > **Q1: Weak baselines, since only GRPO is open-sourced and the other baselines are handmade.**
>
> Thank you for raising this point. We agree that stronger and better-grounded baselines are important for positioning the method fairly. Our intention was not to construct arbitrary baselines, but to instantiate representative token-level credit signals that have already appeared in recent literature, while keeping the RL recipe and compute budget fixed so that only the token-prioritization criterion changes.
>
> To provide a more comprehensive comparison, we include additional strong baselines on Qwen3-8B with 1K context length, including open-sourced generative-PRM method **CAPO [1]** and **ThinkPRM [2]**, value-uncertainty method **AsyPPO [3]**, confidence-oriented **Reweight+Lopti [4]** and **solely reward (instead of reweighting) high-entropy tokens [5]**. Despite the paradigm differences from our specific reweighting scheme, they serve as widely recognized token-level rationales for a rigorous comparison.
>
> |Method|AIME24|AIME25|AMC23|MATH500|Olympiad|Avg|Improvement (vs GRPO)|
> |---|:-:|:-:|:-:|:-:|:-:|:-:|:-:|
> |GRPO|9.3|7.3|59.1|77.1|44.2|39.4|-|
> |+ **CAPO** [1]|10.5|10.9|61.1|78.4|45.9|41.4|+5.08%|
> |+ **ThinkPRM-1.5B** [2]|10.2|10.6|61.0|77.6|45.4|41.0|+4.06%|
> |+ **AsyPPO** [3]|9.6|8.2|59.5|77.8|44.7|40.0|+1.52%|
> |+ **Reweight+Lopti** [4]|8.9|8.8|62.5|78.6|46.5|41.1|+4.31%|
> |+ **high-entropy selection** [5]|10.3|10.5|60.5|77.6|45.6|40.9|+3.81%|
> |+ FlowTracer **(Ours)**|13.0|11.8|65.6|79.7|46.7|43.4|**+10.15%**|
>
> ---
> > **Q2: The improvement is marginal and may be due to random fluctuations.**
>
> We appreciate this concern. We would respectfully argue that the gains are meaningful in the context of RL for reasoning, where different token-level credit strategies often yield relatively small absolute separations under a fixed training budget. In our results, FlowTracer is not winning by an isolated fluctuation on one benchmark; it shows a **consistent advantage across model sizes, context lengths, and task families**, including math reasoning, puzzle solving, and multi-domain QA.
>
> For example, on Qwen3-8B, FlowTracer improves the average math score from **39.4 to 43.4** at 1K and from **50.3 to 52.5** at 8K. On Qwen3-4B, the gain is from **37.1 to 39.4** at 1K and from **44.8 to 48.6** at 8K. On the more challenging long-context setting, the gain on **AIME25** for Qwen3-4B reaches **+5.8** absolute. These are materially larger than the differences between several alternative heuristics in the same tables, which is exactly why we view the flow-based signal as useful.
>
> We agree, however, that variance reporting would strengthen the paper further. Below, we report the average performance and standard derivation across **five random seeds on Qwen3-8B with 1k context length** to to further rule out the potential for random fluctuations. As these multi-seed results demonstrate, **FlowTracer maintains a consistent and clear separation from the baseline variance, confirming that our method's gains are robust and not the result of random fluctuations**. We deeply appreciate your careful suggestion, and will include these confidence statistics in the revised manuscript to provide a more rigorous validation of the improvements.
>
> |Method|AIME24|AIME25|AMC23|MATH|Olympid|Avg|Improvement (vs GRPO)|
> |---|:-:|:-:|:-:|:-:|:-:|:-:|:-:|
> |GRPO|9.2±0.3|7.0±0.6|59.0±0.8|76.9±0.3|45.5±0.5|39.5|-|
> |+ random|8.5±0.3|7.9±0.4|59.5±1.0|77.5±0.9|43.1±0.6|39.3|-0.2(-0.5%)|
> |+ high-entropy|8.9±0.5|8.1±0.3|61.6±0.4|78.2±1.0|46.1±1.3|40.6|+1.1(+2.8%)|
> |+ FlowTracer **(Ours)**|12.7±0.5|11.5±0.5|65.2±0.5|79.3±0.6|46.6±0.3|43.0|+3.5(+9.0%)|
>
>
> ---
>
> We hope this response clarifies that the baselines are literature-motivated and that the observed gains are consistent rather than anecdotal. We appreciate the reviewer’s request for stronger empirical grounding, and we will use it to improve the final version.
>
>
>
> [1] CAPO: Towards Enhancing LLM Reasoning through Generative Credit Assignment. 2025.
>
> [2] Process reward models that think. 2025.
>
> [3] Asymmetric Proximal Policy Optimization: mini-critics boost LLM reasoning. 2025.
>
> [4] Do Not Let Low-Probability Tokens Over-Dominate in RL for LLMs. 2025.
>
> [5] Beyond the 80/20 Rule: High-Entropy Minority Tokens Drive Effective Reinforcement Learning for LLM Reasoning. 2025.

---

> > ### Author Rebuttal · Reviewer_vfJU · 2026-04-02
> >
> > Thanks for the authors' reply.
> >
> > I appreciate the added comparison with more baselines, although the improvement is still too marginal to verify the effectiveness of the proposed method. I would confidently say that, "the gains are meaningful in the context of RL for reasoning" does not make sense. As for what constitutes a significant improvement, please refer to [1][2].
> >
> > I tend to keep the score unchanged until the authors demonstrate the significance of their method.
> >
> > [1] DeepSeekMath: Pushing the Limits of Mathematical Reasoning in Open Language Models. https://arxiv.org/pdf/2402.03300.
> > [2] TTRL: Test-Time Reinforcement Learning. https://arxiv.org/pdf/2504.16084.

---

> > > ### Author Response · Authors · 2026-04-03
> > >
> > > Thank you for your continued engagement and for sharing these references. We appreciate the opportunity to clarify the context of our evaluation and address your concern regarding the scale of the improvements.
> > >
> > > We have carefully considered the results in DeepSeekMath [1] and TTRL [2], and we respectfully believe that **the comparison should be made under the same evaluation regime**. The large gains reported in those works primarily arise from introducing or assessing a new RL paradigm at a broader level, often against models that **do not yet benefit from the same degree of RL refinement**. Our setting is fundamentally different. Our objective is not to measure the leap from things like "no RL" to "with (test-time) RL", but rather to evaluate whether a better token-level credit assignment mechanism can still improve performance within an already mature RL framework.
> > >
> > > This distinction is important because the expected scale of gains is inherently different. Once the backbone RL recipe is already competitive, further improvements are typically much harder to achieve. In this regime, the five recent baselines we added to our evaluation [3,4,5,6,7], including CAPO [3], ThinkPRM [4], and AsyPPO [5], and many other explored methods, are themselves widely recognized as meaningful and valuable contributions to the community. Yet, as shown in our added experiments, baselines' gains over GRPO are **most remaining below 5%** relative improvement, whereas FlowTracer achieves **about 10% relative improvement** under the same setup.
> > >
> > > More importantly, FlowTracer is not proposed merely as another RL improvement. Another core contribution is a **principled flow-based analysis of token importance along the reasoning path**, derived from attention patterns. In this sense, our method provides not only an empirical improvement, but also an analytical perspective: it offers a more grounded way to identify the structural backbone of a reasoning trajectory and to analyze how information is routed from prompt to answer through influential tokens. We believe this perspective is valuable in its own right and may **inspire further investigations into structure-aware RL for long-horizon reasoning**.
> > >
> > > More broadly, while paradigm-shifting breakthroughs are rare, the field also advances through many effective steps that collectively drive sustained progress. We view FlowTracer in this spirit. Our claim is not that FlowTracer should produce the same kind of dramatic absolute jump as introducing RL itself. Rather, our claim is that, in the considerably more challenging regime of improving an already strong RL training pipeline, **consistent gains over recent baselines across model sizes, context lengths, model families, and task families** constitutes strong evidence that the proposed credit assignment signal is effective rather than anecdotal.
> > >
> > > We hope this clarification addresses your concern and helps demonstrate why we believe this work makes a meaningful contribution to the community.
> > >
> > > [1] DeepSeekMath: Pushing the Limits of Mathematical Reasoning in Open Language Models.
> > >
> > > [2] TTRL: Test-Time Reinforcement Learning.
> > >
> > > [3] CAPO: Towards Enhancing LLM Reasoning through Generative Credit Assignment. 2025.
> > >
> > > [4] Process reward models that think. 2025.
> > >
> > > [5] Asymmetric Proximal Policy Optimization: mini-critics boost LLM reasoning. 2025.
> > >
> > > [6] Do Not Let Low-Probability Tokens Over-Dominate in RL for LLMs. 2025.
> > >
> > > [7] Beyond the 80/20 Rule: High-Entropy Minority Tokens Drive Effective Reinforcement Learning for LLM Reasoning. 2025.

---

### Official Review · Reviewer_vVVC · 2026-03-13

**Soundness:** 3
**Presentation:** 2
**Significance:** 2
**Originality:** 3
**Overall Recommendation:** 4
**Confidence:** 4

**Summary:**

The paper addresses the challenge of token-level credit assignment in reinforcement learning (RL) for large language models (LLMs). Standard RL methods typically treat all tokens equally, failing to distinguish critical reasoning steps from fluent filler content. To tackle this issue, the authors propose FlowTracer, an RL framework that models the generation process as an attention-induced directed acyclic graph (DAG). The method applies a Doob-h-like reweighting on the attention graph to enforce local flow conservation and retain only those influence paths that reach the final answer region. Tokens are then scored based on their flow throughput to identify key routing hubs. Finally, these throughput scores are used to non-uniformly weight token-level rewards within a Group Relative Policy Optimization (GRPO) setup. The authors demonstrate consistent performance gains across mathematical reasoning, multi-domain QA, and puzzle-solving benchmarks using Qwen3 and Llama 3 backbones.

**Compliance With Llm Reviewing Policy:**

Affirmed.

**Final Justification:**

I would like to thank the authors for the detailed rebuttal and the additional experimental results provided in the external link.

Specifically, the authors have addressed my primary concern regarding the choice of hard reweighting over continuous schemes. While the design remains primarily empirical, the evidence provided—particularly the ablation studies on flow distribution—demonstrates that the "hard" strategy is a deliberate and effective choice tailored to the observed data characteristics. Furthermore, the additional results sufficiently justify the hyperparameter selections. Given these clarifications and the demonstrated performance gains, I am satisfied with the response.

Accordingly, I have increased my score from 3 to 4.

**Key Questions For Authors:**

1. In Equation 6, the scaling factor $\gamma_{i,t}$ is set to a fixed 1.5 for the top 40% of tokens. Since the flow throughput $\tau(k)$ provides a precise, continuous measure of importance, why did the authors opt for a hard threshold rather than continuously scaling the reward by the normalized flow score?
2. The ablation study shows that Top-40% token selection yields the best performance. Have the authors experimented with dynamic thresholds (e.g., selecting tokens that account for a cumulative percentage of the total flow) rather than a fixed top-k percentage?
3. The authors average attention maps specifically from the "middle layers" (ranging from $L/3$ to $2L/3$). The ablation study indicates that this was chosen simply because it "consistently yields the best performance," while using all layers dilutes the signal and leads to worse performance. If the flow theory is universally sound, why does it perform poorly when applied to the entire network? This lack of mechanistic justification significantly weakens the robustness of the method.

**Limitations:**

The authors include an Impact Statement and briefly mention leaving finer-grained mechanistic interpretation to future work but do not include a dedicated "Limitations" section.
Suggestion for authors: Please include a clear limitations section discussing potential failure modes.

**Strengths And Weaknesses:**

# Strengths
1. The mathematical formulation is rigorous; the use of a Doob-h-like transform effectively resolves the local flow conservation issue inherent in raw attention maps. Furthermore, the authors do not merely assume these high-flow tokens are important; they validate this through a causal intervention experiment on the GSM8K dataset. The experiment shows that masking high-flow tokens leads to a 45.9% answer change rate, compared to just 14.9% for low-flow tokens, thereby proving their structural importance. The evaluation is also thorough, covering multiple context lengths (1K and 8K).
2. The submission is well-structured and highly readable. The visualizations are excellent; Figure 1 clearly illustrates the theoretical pipeline, and Figure 2 effectively grounds the theory by visualizing token flow importance on a real math problem. The narrative logically flows from motivation to graph theory and finally to RL integration.
3. Token-level credit assignment is a well-known and difficult bottleneck in Reinforcement Learning with Verifiable Rewards (RLVR). By providing a structured way to assign fine-grained credit with a claimed computational overhead of only 2.2% to 4.5%, this paper offers high practical utility for the community. The empirical improvements are substantial, notably achieving a +5.8 absolute accuracy gain on AIME25 using the Qwen3-4B model at an 8K context length.
4. Shifting the perspective to global, structure-aware information flow on an attention-induced DAG is a highly creative and novel direction for RL credit assignment in LLMs.
# Weaknesses
1. The authors explicitly acknowledge that a Transformer state $H^{(l)}$ is updated via both the attention output and the Feed-Forward Network (FFN) with residual connections. Yet, the flow calculation completely ignores the MLPs (i.e., FFNs), which contain the majority of the model's parameters and are crucial for non-linear information routing.
2. The NLP community has long debated whether "attention is explanation." By treating raw attention weights as direct "influence couplings," the authors bypass well-documented phenomena such as attention collapse or the fact that models often attend to uninformative structural tokens (e.g., punctuation) as a default state.
3. While the method shows strong gains on Qwen-based models, the absolute performance improvements in the Llama family (e.g., Llama-3.1-8B and Llama-3.2-3B) are modest (+1.4% and +1.1% on average, respectively). This discrepancy raises questions about how universally significant improvements are across diverse model families that might not have the same baseline reasoning capabilities or attention characteristics.
4. To implement the credit assignment, the authors simply apply a hardcoded multiplier ($\gamma_{flow}=1.5$) to the GRPO loss for the top 40% of tokens. This is not a mathematically rigorous derivation of token advantage; rather, it is a binary step-function heuristic grafted onto the existing GRPO formula. Why 1.5? Why use a hard cutoff at 40% instead of a continuous weighting scheme?
5. While the application of the Doob-h-like transform to enforce local flow conservation is a clever graph-theoretic trick, the broader idea of using internal metrics (like entropy, gradients, or attention itself) for credit assignment has already been explored in recent concurrent work. The core novelty is limited and heavily reliant on unproven graph construction.

---

> ### Author Rebuttal · Authors · 2026-03-31
>
> ---
> > **Q1: Flow is computed only from attention maps, while FFNs also route important information.**
>
> We agree, but our method does not claim to capture the full internal computation or to reconstruct every internal pathway. Instead, **our goal is to extract a practical signal for credit assignment**. We focus on attention because it provides an explicit token-to-token interaction structure, enabling principled analysis of multi-hop routing (as evidenced by intervention results and downstream RL gains). We will clarify this scope and add a dedicated discussion of FFN-related limitations in the revised manuscript.
>
> ---
> > **Q2: The explanatory power of attention is debated; raw attention may reflect artifacts.**
>
> We agree, and we do not claim that "attention is explanation" in a causal or exhaustive sense. Our claim is narrower: raw attention contains both structure and noise, and FlowTracer filters it into an answer‑targeted, flow‑conserving backbone more useful for analysis and RL credit assignment than raw weights alone. For example, the Doob‑h‑like conditioning removes non‑answer‑reaching branches, and the throughput score highlights tokens that repeatedly mediate answer‑relevant paths.
>
> Empirically, this signal is not arbitrary: perturbing high‑flow tokens changes answers far more often than low‑flow ones (Table 1), and RL reweighting based on these tokens consistently outperforms GRPO and attention‑only baselines (Tables 2–4).
>
> ---
> > **Q3: Improvements on the Llama family seem modest, which may limit universality.**
>
> Thanks. Indeed, absolute gains should be interpreted in the context of the underlying backbone strength. For **Llama-3.1-8B**, the average score improves from **7.7** under GRPO to **9.1** with FlowTracer (**~18% relative improvement**); for **Llama-3.2-3B**, from **4.8** to **5.9** (**~23% relative improvement**). More broadly, FlowTracer consistently improves over GRPO and other heuristics on both Qwen and Llama, suggesting the signal is not architecture‑specific. We agree that the gain magnitude may depend on factors such as baseline reasoning ability, and we will discuss this more explicitly.
>
> ---
> > **Q4: Why not continuous reweighting?**
>
> This choice is empirical rather than theoretically unique. Flow throughput is highly skewed: a small subset of tokens carry most flow, while many have very small values. Continuous weighting can be unstable or let outliers dominate; a moderate binary emphasis is more robust. We experiment with continuous reweighting and soft (cumulative‑flow) thresholds, finding both less effective than the hard Top‑40% design. Such a design may need more detailed designs in the future.
>
> Key observations (**the tables are in https://anonymous.4open.science/r/A508C/Rebuttal_to_Reviewer_vVVC.pdf** on Qwen3‑4B with 1K length) include:
>
> 1. **Hard vs. continuous reweighting (Table.1 within the link):** Continuous schemes (raw flow, tanh+zscore, sigmoid, MAD, log1p) generally underperform hard reweighting. The optimal multiplier is around 1.5 (values 1.2–1.8 work; <1.0 gives negative optimization, >1.8 tends to crash).
>
> 2. **Hard vs. Soft (Dynamic) Thresholds  (Table.2 within the link):** Soft thresholds (selecting tokens accounting for a cumulative percentage of total flow) also underperform the hard 40% threshold. Because flow is concentrated in a tiny subset, low soft ratios often pick only outliers, while high ratios include plenty of low‑flow tokens, making it difficult to select a stable and effective threshold.
>
> ---
> > **Q5. Novelty compared to other credit assignment methods with internal metrics (entropy, gradient or attention)**
>
> We appreciate this concern. Prior methods largely use point‑wise internal signals (e.g., uncertainty, gradient). In contrast, we explicitly model **global, answer‑targeted multi‑hop flow** on a token DAG, combined with a Doob‑h‑like transform for local conservation and pruning. Our contribution is two‑fold: (1) a structured analysis perspective on answer‑routed information, and (2) an RL integration that leverages this signal for credit assignment. We will sharpen this positioning and contrast with concurrent approaches.
>
> ---
> > **Q6: Why do middle-layer attention maps work best?**
>
> The flow formalism is general, but the quality of the extracted reasoning backbone still depends on the attention source used to build the graph. Early layers are closer to lexical and prompt-local processing, while late layers are often more tied to token emission and answer formatting. For intermediate layers, contextual integration and multi-step dependency routing are more prominent. If early and late layers contain interactions less relevant to reasoning backbones, mixing them with the middle-layer structure reduces contrast.
>
> ---
>
> We hope this response helps clarify the scope, motivation, and empirical support of our approach. We are grateful for the reviewer’s comments, which will help us strengthen both the framing and the analysis of the paper.

---

> > ### Author Rebuttal · Reviewer_vVVC · 2026-04-03
> >
> > I would like to thank the authors for the detailed rebuttal and the additional experimental results provided in the external link.
> >
> > Specifically, the authors have addressed my primary concern regarding the choice of hard reweighting over continuous schemes. While the design remains primarily empirical, the evidence provided—particularly the ablation studies on flow distribution—demonstrates that the "hard" strategy is a deliberate and effective choice tailored to the observed data characteristics. Furthermore, the additional results sufficiently justify the hyperparameter selections. Given these clarifications and the demonstrated performance gains, I am satisfied with the response.
> >
> > Accordingly, I have increased my score from 3 to 4.

---

> > > ### Author Response · Authors · 2026-04-07
> > >
> > > Thank you for your valuable feedback and your decision to increase the score. It is highly encouraging for us to know that the additional experimental results resolved your concerns, and we are completely willing to address any further questions.

---

### Decision · Program_Chairs · 2026-04-30

**Decision:**

Accept (regular)

**Comment:**

The paper studies token-level credit assignment in RL for LLM reasoning. The main idea is to trace how information flows from the prompt to the final answer through the model’s attention patterns. Based on this, the paper proposes FlowTracer, which builds an answer-targeted flow graph over tokens, identifies “high-throughput” tokens as part of a reasoning backbone, and uses them to reweight GRPO updates. Experiments show consistent gains across math reasoning, QA, and puzzle tasks, with especially strong improvements in harder or longer-context settings.

Reviewers agreed that the paper tackles an important problem and appreciate the novelty. They found the graph-based view of reasoning flow interesting, and several reviewers liked that the method is lightweight, easy to plug into GRPO, and supported by broad experiments. The causal intervention results and the low overhead were also seen as strengths.

The main weaknesses were about interpretation and significance. Some reviewers questioned the premise of using attention as a proxy for influence, and noted that the method ignores FFN contributions. Others wanted stronger discussion of whether the method may over-penalize locally useful tokens inside globally wrong trajectories, and whether the answer-region definition or fixed token-selection rule makes the method brittle.

The rebuttal addressed many of these concerns. The authors added stronger baseline comparisons, multi-seed results, and extra ablations on hard versus continuous weighting, soft thresholds, structural versus semantic tokens, and answer-region robustness. Some conceptual limitations remain, especially around attention-based interpretation and incomplete token-level supervision, but overall the rebuttal substantially strengthened the paper.